# Optimal Multiclass U-Calibration Error and Beyond

**Haipeng Luo**[*]
University of Southern California
`haipengl@usc.edu`

**Spandan Senapati**[*]
University of Southern California
`ssenapat@usc.edu`

**Vatsal Sharan**[*]
University of Southern California
`vsharan@usc.edu`

## Abstract

We consider the problem of *online multiclass U-calibration*, where a forecaster aims to make sequential distributional predictions over $K$ classes with low *U-calibration error*, that is, low regret with respect to *all* bounded proper losses simultaneously. Kleinberg et al. (2023) developed an algorithm with U-calibration error $\mathcal{O}(K\sqrt{T})$ after $T$ rounds and raised the open question of what the optimal bound is. We resolve this question by showing that the optimal U-calibration error is $\Theta(\sqrt{KT})$ — we start with a simple observation that the Follow-the-Perturbed-Leader algorithm of Daskalakis and Syrgkanis (2016) achieves this upper bound, followed by a matching lower bound constructed with a specific proper loss (which, as a side result, also proves the optimality of the algorithm of Daskalakis and Syrgkanis (2016) in the context of online learning against an adversary with finite choices). We also strengthen our results under natural assumptions on the loss functions, including $\Theta(\log T)$ U-calibration error for Lipschitz proper losses, $\mathcal{O}(\log T)$ U-calibration error for a certain class of decomposable proper losses, U-calibration error bounds for proper losses with a low covering number, and others.

## 1 Introduction

We consider the fundamental problem of making sequential probabilistic predictions over an outcome (e.g., predicting the probability of tomorrow's weather being sunny, cloudy, or rainy). Specifically, at each time $t = 1, \ldots, T$, a forecaster/learner predicts $\boldsymbol{p}_t \in \Delta_K$, where $\Delta_K$ denotes the probability simplex over $K$ outcomes. At the same time, an adversary decides the true outcome, encoded by a one-hot vector $\boldsymbol{y}_t \in \mathcal{E} := \{\boldsymbol{e}_1, \ldots, \boldsymbol{e}_K\}$, where $\boldsymbol{e}_i$ denotes the $i$-th standard basis vector of $\mathbb{R}^K$. The forecaster observes $\boldsymbol{y}_t$ at the end of time $t$.

A popular approach to measure the performance of a forecaster is to measure her *regret* against the best fixed prediction in hindsight. Fixing some loss function $\ell : \Delta_K \times \mathcal{E} \to \mathbb{R}$, the regret of the forecaster's predictions with respect to $\ell$ is defined as $\text{REG}_\ell := \sum_{t=1}^{T} \ell(\boldsymbol{p}_t, \boldsymbol{y}_t) - \inf_{\boldsymbol{p} \in \Delta_K} \sum_{t=1}^{T} \ell(\boldsymbol{p}, \boldsymbol{y}_t)$. Perhaps the most common class of loss functions to evaluate a forecaster are *proper* loss functions. A loss function is proper if $\mathbb{E}_{\boldsymbol{y} \sim \boldsymbol{p}}[\ell(\boldsymbol{p}, \boldsymbol{y})] \leq \mathbb{E}_{\boldsymbol{y} \sim \boldsymbol{p}}[\ell(\boldsymbol{p}', \boldsymbol{y})]$ for all $\boldsymbol{p}, \boldsymbol{p}' \in \Delta_K$. Hence proper loss functions incentivize the forecaster to predict the true probability of the outcome (to the best of their knowledge). We will focus on proper loss functions in this work.

Note, however, that regret is measured with respect to a specific loss function $\ell$. It is unclear which proper loss one should minimize over for the specific application at hand — and there could even

---

[*]Author ordering is alphabetical.

38th Conference on Neural Information Processing Systems (NeurIPS 2024).

be multiple applications with different loss functions which use the forecasters's prediction. Could it be possible for a forecaster to simultaneously enjoy low regret with respect to all proper loss functions? This questions was raised in the interesting recent work of Kleinberg et al. (2023). They propose the notion of *U-calibration error* $\mathsf{UCal}_{\mathcal{L}} := \mathbb{E}\left[\sup_{\ell \in \mathcal{L}} \mathrm{REG}_\ell\right]$ (and a weaker version *pseudo U-calibration error* $\mathsf{PUCal}_{\mathcal{L}} := \sup_{\ell \in \mathcal{L}} \mathbb{E}[\mathrm{REG}_\ell]$) for a family of proper loss functions $\mathcal{L}$. A forecaster with low U-calibration error thus enjoys good performance with respect to *all* loss functions in $\mathcal{L}$ simultaneously. Unless explicitly mentioned, we shall let $\mathcal{L}$ denote the set of all bounded (in $[-1, 1]$) proper losses (as in Kleinberg et al. (2023)), and drop the subscript in $\mathsf{UCal}_{\mathcal{L}}$ and $\mathsf{PUCal}_{\mathcal{L}}$ for convenience.

The simplest way to get low U-calibration error is via the classical notion of low *calibration error* (Dawid, 1982), defined as $\mathsf{Cal} := \sum_{\boldsymbol{p} \in \Delta_K} \left\| \sum_{t; \boldsymbol{p}_t = \boldsymbol{p}} (\boldsymbol{p} - \boldsymbol{y}_t) \right\|_1$. Intuitively, a forecaster with low calibration error guarantees that whenever she makes a prediction $\boldsymbol{p}$, the empirical distribution of the true outcome is indeed close to $\boldsymbol{p}$. Kleinberg et al. (2023) prove that $\mathsf{PUCal} \leq \mathsf{UCal} = \mathcal{O}(\mathsf{Cal})$ and thus a well-calibrated forecaster must have small U-calibration error. However, getting low calibration error is difficult and faces known barriers: the best existing upper bound on $\mathsf{Cal}$ is $\mathcal{O}(T^{\frac{K}{K+1}})$ (Blum et al., 2008), and there is a $\Omega(T^{0.528})$ lower bound (for $K = 2$) (Qiao and Valiant, 2021). Therefore, a natural question to ask is if it is possible to side-step calibration and directly get low U-calibration error. Kleinberg et al. (2023) answer this in the affirmative, and show that there exist simple and efficient algorithms with $\mathsf{UCal} = \mathcal{O}(\sqrt{T})$ for $K = 2$ and $\mathsf{PUCal} = \mathcal{O}(K\sqrt{T})$ for general $K$. This provides a strong decision-theoretic motivation for considering U-calibration error as opposed to calibration error; we refer the reader to Kleinberg et al. (2023) for further discussion.

Following up Kleinberg et al. (2023), this paper addresses the following question that was left open in their work: *"What is the minimax optimal multiclass U-calibration error?"* We give a complete answer to this question (regarding $\mathsf{PUCal}$) by showing matching upper and lower bounds. Moreover, we identify several broad sub-classes of proper losses for which much smaller U-calibration error is possible. Concretely, our contributions are as follows.

## 1.1 Contributions and Technical Overview

First, we show that the minimax optimal value of $\mathsf{PUCal}$ is $\Theta(\sqrt{KT})$:

- In Section 3.1, we start by showing that a simple modification to the noise distribution of the Follow-the-Perturbed-Leader (FTPL) algorithm of Kleinberg et al. (2023) improves their $\mathsf{PUCal} = \mathcal{O}(K\sqrt{T})$ bound to $\mathcal{O}(\sqrt{KT})$. In fact, our algorithm coincides with that of Daskalakis and Syrgkanis (2016) designed for an online learning setting with a fixed loss function and an adversary with only finite choices. The reason that it works for any proper losses simultaneously in our problem is because for any set of outcomes, the empirical risk minimizer with respect to any proper loss is always the average of the outcomes (*c.f.* property (1)).

- We then show in Section 3.2 that there exists one particular proper loss $\ell$ such that any algorithm has to suffer $\mathrm{REG}_\ell = \Omega(\sqrt{KT})$ in the worst case, hence implying $\mathsf{PUCal} = \Omega(\sqrt{KT})$. While our proof follows a standard randomized argument, the novelty lies in the construction of the proper loss and the use of an anti-concentration inequality to bound the expected loss of the benchmark. We remark that, as a side result, our lower bound also implies the optimality of the FTPL algorithm of Daskalakis and Syrgkanis (2016) in their setting, which is unknown before to our knowledge.

While Kleinberg et al. (2023) only consider $\mathsf{PUCal}$ for general $K$, we take a step forward and further study the stronger measure $\mathsf{UCal}$ (recall $\mathsf{PUCal} \leq \mathsf{UCal}$). We start by showing an upper bound on $\mathsf{UCal}_{\mathcal{L}'}$ for the same FTPL algorithm and for any loss class $\mathcal{L}'$ with a finite covering number. Then, we consider an even simpler algorithm, Follow-the-Leader (FTL), which is deterministic and makes $\mathsf{UCal}$ and $\mathsf{PUCal}$ trivially equal, and identify two broad classes of loss functions where FTL achieves logarithmic U-calibration error, an exponential improvement over the worst case:

- In Section 4.1, we show that for the class $\mathcal{L}_G$ of $G$-Lipschitz bounded proper losses (which includes standard losses such as the squared loss and spherical loss), FTL ensures $\mathsf{PUCal}_{\mathcal{L}_G} = \mathsf{UCal}_{\mathcal{L}_G} = \mathcal{O}(G \log T)$.

  We further show that all algorithms must suffer $\mathsf{PUCal}_{\mathcal{L}_G} = \Omega(\log T)$. While we prove this lower bound using the standard squared loss that is known to admit $\Theta(\log T)$ regret in many online

learning settings (e.g., Abernethy et al. (2008)), to our knowledge it has not been studied in our setting where the learner's decision set is a simplex and the adversary has finite choices. Indeed, our proof is also substantially different from Abernethy et al. (2008) and is one of the most technical contributions of our work.

- Next, in Section 4.2, we identify a class $\mathcal{L}_{dec}$ of losses that are decomposable over the $K$ outcomes and additionally satisfy some mild regularity conditions, and show that FTL again achieves $\mathsf{PUCal} = \mathsf{UCal} = \mathcal{O}(\log T)$ (ignoring other dependence). This class includes losses induced by a certain family of Tsallis entropy that are not Lipschitz. The key idea of our proof is to show that even though the loss might not be Lipschitz, its gradient grows at a controlled rate.

- Given these positive results on FTL, one might wonder whether FTL is generally a good algorithm for any proper losses. We answer this question in the negative in Section 4.3 by showing that there exists a bounded proper loss such that the regret of FTL is $\Omega(T)$. This highlights the need of using FTPL if one cares about all proper losses (or at least losses not in $\mathcal{L}_G$ or $\mathcal{L}_{dec}$).

## 1.2 Related Work

For calibration, Foster and Vohra (1998) proposed the first algorithm for the binary setting with an (expected) $\mathcal{O}(T^{\frac{2}{3}})$ calibration error (see also Blum and Mansour (2007) and Hart (2022) for a different proof of the result). In the multiclass setting, Blum et al. (2008) have shown an $\mathcal{O}(T^{\frac{K}{K+1}})$ calibration error. Several works have studied other variants of calibration error, such as the most recently proposed Distance to Calibration (Błasiok et al., 2023; Qiao and Zheng, 2024; Arunachaleswaran et al., 2024); see the references therein for other earlier variants.

A recent research trend, initiated by Gopalan et al. (2022), has centered around the concept of simultaneous loss minimization, also known as *omniprediction*. Garg et al. (2024) study an online adversarial version of it, and U-calibration can be seen as a special non-contextual case of their setting with only proper losses considered. Their results, however, are not applicable here due to multiple reasons: for example, they consider only the binary case ($K = 2$), and their algorithm is either only designed for Lipschitz convex loss functions or computationally inefficient. We also note that omniprediction has been shown to have a surprising connection with multicalibration (Hébert-Johnson et al., 2018), a multi-group fairness notion, making it an increasingly important topic (Gopalan et al., 2022; Błasiok et al., 2024; Gopalan et al., 2023a,b).

## 2 Preliminaries

**Notation:** We use lowercase bold alphabets to denote vectors. $\mathbb{N}$, $\mathbb{N}_{\geq 0}$ denote the set of positive, non-negative integers respectively. For any $m \in \mathbb{N}$, $[m]$ denotes the index set $\{1, \ldots, m\}$. We use $\Delta_K$ to denote the $(K-1)$-dimensional simplex, i.e., $\Delta_K := \{\boldsymbol{p} \in \mathbb{R}^K \mid p_i \geq 0, \sum_{i=1}^{K} p_i = 1\}$. The $i$-th standard basis vector (dimension inferred from the context) is denoted by $\boldsymbol{e}_i$, and we use $\mathcal{E}$ to represent the set $\{\boldsymbol{e}_1, \ldots, \boldsymbol{e}_K\}$ of all basis vectors of $\mathbb{R}^K$. By default, $\|\cdot\|$ denotes the $\ell_2$ norm.

**Proper Losses:** Throughout the paper, we consider the class of bounded proper losses $\mathcal{L} := \{\ell : \Delta_K \times \mathcal{E} \to [-1, 1] \mid \ell \text{ is proper}\}$ or a subset of it. We emphasize that convexity (in the first argument) is never needed in our results. As mentioned, a loss $\ell$ is proper if predicting the true distribution from which the outcome is sampled from gives the smallest loss in expectation, that is, $\mathbb{E}_{\boldsymbol{y} \sim \boldsymbol{p}}[\ell(\boldsymbol{p}, \boldsymbol{y})] \leq \mathbb{E}_{\boldsymbol{y} \sim \boldsymbol{p}}[\ell(\boldsymbol{p}', \boldsymbol{y})]$ for all $\boldsymbol{p}, \boldsymbol{p}' \in \Delta_K$.

For a proper loss $\ell$, we refer to $\ell(\boldsymbol{p}, \boldsymbol{y})$ as its *bivariate* form. The *univariate* form of $\ell$ is defined as $\ell(\boldsymbol{p}) := \mathbb{E}_{\boldsymbol{y} \sim \boldsymbol{p}}[\ell(\boldsymbol{p}, \boldsymbol{y})]$. It turns out that a loss is proper only if its univariate form is concave. Moreover, one can construct a proper loss using a concave univariate form based on the following characterization lemma.

**Lemma 1** (Theorem 2 of Gneiting and Raftery (2007)). *A loss $\ell : \Delta_K \times \mathcal{E} \to \mathbb{R}$ is proper if and only if there exists a concave function $f$ such that $\ell(\boldsymbol{p}, \boldsymbol{y}) = f(\boldsymbol{p}) + \langle \boldsymbol{g_p}, \boldsymbol{y} - \boldsymbol{p} \rangle$ for all $\boldsymbol{p} \in \Delta_K, \boldsymbol{y} \in \mathcal{E}$, where $\boldsymbol{g_p}$ denotes a subgradient of $f$ at $\boldsymbol{p}$. Also, $f$ is the univariate form of $\ell$.*

We provide several examples of proper losses below:

- The spherical loss is $\ell(\boldsymbol{p}, \boldsymbol{y}) = -\frac{\langle \boldsymbol{p}, \boldsymbol{y} \rangle}{\|\boldsymbol{p}\|}$, which is $\sqrt{K}$-Lipschitz (Proposition B.1) but *non-convex* in $\boldsymbol{p}$. Its univariate form is $\ell(\boldsymbol{p}) = -\|\boldsymbol{p}\|$.

- The squared loss (also known as the Brier score) is $\ell(\boldsymbol{p}, \boldsymbol{y}) = \frac{1}{2}\|\boldsymbol{p} - \boldsymbol{y}\|^2$, which is clearly 2-Lipschitz and convex in $\boldsymbol{p}$. Its univariate form is $\ell(\boldsymbol{p}) = 1 - \|\boldsymbol{p}\|^2$.

- Generalizing the squared loss, we consider the univariate form $\ell(\boldsymbol{p}) = -\tilde{c}_K \sum_{i=1}^{K} p_i^\alpha$ for $\alpha > 1$ and some constant $\tilde{c}_K > 0$, which is the Tsallis entropy and is concave. The induced proper loss is $\ell(\boldsymbol{p}, \boldsymbol{y}) = \tilde{c}_K(\alpha - 1) \sum_{i=1}^{K} p_i^\alpha - \tilde{c}_K \alpha \sum_{i=1}^{K} p_i^{\alpha-1} y_i$, which is *not Lipschitz* for $\alpha \in (1, 2)$.[*]

The following fact is critical for U-calibration: for any $n \in \mathbb{N}$ and a sequence of outcomes $\boldsymbol{y}_1, \ldots, \boldsymbol{y}_n \in \mathcal{E}$, the *mean forecaster* is always the empirical risk minimizer for any proper loss $\ell$: (the proof is by definition and included in Appendix B for completeness):

$$\frac{1}{n} \sum_{j=1}^{n} \boldsymbol{y}_j \in \operatorname*{argmin}_{p \in \Delta_K} \frac{1}{n} \sum_{j=1}^{n} \ell(\boldsymbol{p}, \boldsymbol{y}_j). \tag{1}$$

**Problem Setting:** As mentioned, the problem we study follows the following protocol: at each time $t = 1, \ldots, T$, a forecaster predicts a distribution $\boldsymbol{p}_t \in \Delta_K$ over $K$ possible outcomes, and at the same time, an adversary decides the true outcome encoded by a one-hot vector $\boldsymbol{y}_t \in \mathcal{E}$, which is revealed to the forecaster at the end of time $t$.

For a fixed proper loss function $\ell$, the regret of the forecaster is defined as: $\mathrm{REG}_\ell := \sum_{t=1}^{T} \ell(\boldsymbol{p}_t, \boldsymbol{y}_t) - \inf_{\boldsymbol{p} \in \Delta_K} \sum_{t=1}^{T} \ell(\boldsymbol{p}, \boldsymbol{y}_t)$, which, according to property (1), can be written as $\sum_{t=1}^{T} \ell(\boldsymbol{p}_t, \boldsymbol{y}_t) - \sum_{t=1}^{T} \ell(\boldsymbol{\beta}, \boldsymbol{y}_t)$ where $\boldsymbol{\beta} := \frac{1}{T} \sum_{t=1}^{T} \boldsymbol{y}_t$ is simply the empirical average of all outcomes.

Our goal is to ensure low regret against a class of proper losses simultaneously. We define U-calibration error as $\mathsf{UCal}_\mathcal{L} = \mathbb{E}\left[\sup_{\ell \in \mathcal{L}} \mathrm{REG}_\ell\right]$ and pseudo U-calibration error as $\mathsf{PUCal}_\mathcal{L} = \sup_{\ell \in \mathcal{L}} \mathbb{E}[\mathrm{REG}_\ell]$ for a family of loss functions $\mathcal{L}$. Unless explicitly mentioned, $\mathcal{L}$ denotes the set of all bounded (in $[-1, 1]$) proper losses and is dropped from the subscripts for convenience.

**Oblivious Adversary versus Adaptive Adversary:** As is standard in online learning, an oblivious adversary decides all outcomes $\boldsymbol{y}_1, \ldots, \boldsymbol{y}_T$ ahead of the time with the knowledge of the forecaster's algorithm (but not her random seeds), while an adaptive adversary decides each $\boldsymbol{y}_t$ with the knowledge of past forecasts $\boldsymbol{p}_1, \ldots, \boldsymbol{p}_{t-1}$. Except for one result (upper bound on $\mathsf{UCal}$ for a class with low-covering number), all our upper bounds hold for the stronger adaptive adversary, and all our lower bounds hold for the weaker oblivious adversary (which makes the lower bounds stronger).

## 3 Optimal U-calibration Error

In this section, we prove that the minimax optimal pseudo U-calibration error is $\Theta(\sqrt{KT})$.

### 3.1 Algorithm

As mentioned, our algorithm makes a simple change to the noise distribution of the FTPL algorithm of Kleinberg et al. (2023) and in fact coincides with the algorithm of Daskalakis and Syrgkanis (2016) designed for a different setting. To this end, we start by reviewing their setting and algorithm. Specifically, Daskalakis and Syrgkanis (2016) consider the following online learning problem: at each time $t \in [T]$, a learner chooses an action $\boldsymbol{a}_t \in \mathcal{A}$ for some action set $\mathcal{A}$; at the same time, an adversary selects an outcome $\boldsymbol{\theta}_t$ from a finite set $\Theta := \{\hat{\boldsymbol{\theta}}_1, \ldots, \hat{\boldsymbol{\theta}}_K\}$ of size $K$; finally, the learner observes $\boldsymbol{\theta}_t$ and incurs loss $h(\boldsymbol{a}_t, \boldsymbol{\theta}_t)$ for some arbitrary loss function $h : \mathcal{A} \times \Theta \to [-1, 1]$ that is fixed and known to the learner. Daskalakis and Syrgkanis (2016) propose the following FTPL algorithm: at each time $t$, randomly generate a set of hallucinated outcomes, where the number of each possible outcome $\hat{\boldsymbol{\theta}}_i$ for $i \in [K]$ follows independently a geometric distribution with parameter $\sqrt{K/T}$, and then output the empirical risk minimizer using both the true outcomes and the hallucinated outcomes as the final action $\boldsymbol{a}_t$. Formally, $\boldsymbol{a}_t \in \operatorname*{argmin}_{\boldsymbol{a} \in \mathcal{A}} \sum_{i=1}^{K} Y_{t,i} h(\boldsymbol{a}, \hat{\boldsymbol{\theta}}_i)$ where $Y_{t,i} = |\{s < t \mid \boldsymbol{\theta}_s = \hat{\boldsymbol{\theta}}_i\}| + m_{t,i}$

---

[*]Here, the scaling constant $\tilde{c}_K$ is such that $\ell(\boldsymbol{p}, \boldsymbol{y}) \in [-1, 1]$.

---
**Algorithm 1** FTPL with geometric noise for U-calibration
---
1: **for** $t = 1, \ldots, T$
2:  For each $i \in [K]$, sample $m_{t,i}$ independently from a geometric distribution with parameter $\sqrt{K/T}$, and compute $Y_{t,i} = |\{s < t \mid \boldsymbol{y}_s = \boldsymbol{e}_i\}| + m_{t,i}$.
3:  Predict $\boldsymbol{p}_t \in \Delta_K$ such that $p_{t,i} = Y_{t,i} / \sum_{k=1}^K Y_{t,k}$, and observe the true outcome $\boldsymbol{y}_t \in \mathcal{E}$.
4: **end for**
---

and $m_{t,i}$ is an i.i.d. sample of a geometric distribution with parameter $\sqrt{K/T}$. This simple algorithm enjoys the following regret guarantee.

**Theorem 1** (Appendix F.3 of Daskalakis and Syrgkanis (2016))**.** *The FTPL algorithm described above satisfies the following regret bound:* $\mathbb{E}\left[\sum_{t=1}^T h(\boldsymbol{a}_t, \boldsymbol{\theta}_t) - \inf_{\boldsymbol{a} \in \mathcal{A}} \sum_{t=1}^T h(\boldsymbol{a}, \boldsymbol{\theta}_t)\right] \leq 4\sqrt{KT}$, *where the expectation is taken over the randomness of both the algorithm and the adversary.*

Now we are ready to discuss how to apply their algorithm to our multiclass U-calibration problem. Naturally, we take $\mathcal{A} = \Delta_K$ and $\Theta = \mathcal{E}$. What $h$ should we use when we care about all proper losses? It turns out that this does not matter (an observation made by Kleinberg et al. (2023) already): according to property (1), the mean forecaster taking into account both the true outcomes and the hallucinated ones (that is, $p_{t,i} = Y_{t,i} / \sum_{k=1}^K Y_{t,k}$) is a solution of $\operatorname{argmin}_{\boldsymbol{p} \in \Delta_K} \sum_{i=1}^K Y_{t,i} h(\boldsymbol{p}, \boldsymbol{e}_i)$ for *any* proper loss $h \in \mathcal{L}$! This immediately leads to the following result.

**Corollary 1.** *Algorithm 1 ensures* $\mathsf{PUCal} \leq 4\sqrt{KT}$ *against any adaptive adversary.*

We remark that the only difference of Algorithm 1 compared to that of Kleinberg et al. (2023) is that $m_{t,i}$ is sampled from a geometric distribution instead of a uniform distribution in $\{0, 1, \ldots, \lfloor\sqrt{T}\rfloor\}$. Using such noises that are skewed towards smaller values leads to better trade-off between the stability of the algorithm and the expected noise range, which is the key to improve the regret bound from $\mathcal{O}(K\sqrt{T})$ to $\mathcal{O}(\sqrt{KT})$.

## 3.2 Lower Bound

We now complement the upper bound of the previous section with a matching lower bound. Similar to the Multi-Armed Bandit problem, the regime of interest is $T = \Omega(K)$.

**Theorem 2.** *There exists a proper loss $\ell$ with range $[-1, 1]$ such that the following holds: for any online algorithm* ALG*, there exists a choice of $\boldsymbol{y}_1, \ldots, \boldsymbol{y}_T$ by an oblivious adversary such that the expected regret $\mathbb{E}[\mathrm{REG}_\ell]$ of* ALG *is $\Omega(\sqrt{KT})$ when $T \geq 12K$.*

We defer to the proof to Appendix C and highlight the key ideas and novelty here. First, the proper loss we use to prove the lower bound takes the following univariate form $\ell(\boldsymbol{p}) = -\frac{1}{2} \sum_{i=1}^K \left|p_i - \frac{1}{K}\right|$, which is in fact a direct generalization of the so-called "V-shaped loss" studied in Kleinberg et al. (2023) for the binary case. More specifically, they show that in the binary case, V-shaped losses are the "hardest" in the sense that low regret with respect to all V-shaped losses directly implies low regret with respect to all proper losses (that is, low U-calibration error). On the other hand, they also prove that this is *not true* for the general multiclass case. Despite this fact, here, we show that V-shaped loss is still the "hardest" in the multiclass case in a different sense: it is the hardest loss for any algorithm with $\mathsf{PUCal} = \mathcal{O}(\sqrt{KT})$.

With this loss function, we then follow a standard probabilistic argument and consider a randomized oblivious adversary that samples $\boldsymbol{y}_1, \ldots, \boldsymbol{y}_T$ i.i.d. from the uniform distribution over $\mathcal{E}$. For such an adversary, we argue the following: (a) the expected loss incurred by ALG is non-negative, i.e., $\mathbb{E}\left[\sum_{t=1}^T \ell(\boldsymbol{p}_t, \boldsymbol{y}_t)\right] \geq 0$, where the expectation is taken over $\boldsymbol{y}_1, \ldots, \boldsymbol{y}_T$ and any internal randomness in ALG; (b) the expected loss incurred by the benchmark is bounded as $\mathbb{E}\left[\inf_{\boldsymbol{p} \in \Delta_K} \sum_{t=1}^T \ell(\boldsymbol{p}, \boldsymbol{y}_t)\right] \leq -c\sqrt{KT}$ for some universal positive constant $c$, where the expectation is over $\boldsymbol{y}_1, \ldots, \boldsymbol{y}_T$. Together, this implies that the expected regret of ALG is at least $c\sqrt{KT}$ in this randomized environment, which further implies that there must exist one particular sequence of $\boldsymbol{y}_1, \ldots, \boldsymbol{y}_T$ such that the expected

regret of ALG is at least $c\sqrt{KT}$, finishing the proof. We remark that our proof for (b) is novel and based on an anti-concentration inequality for Bernoulli random variables (Lemma A.4).

We discuss some immediate implications of Theorem 2 below. First, it implies that in the online learning setting of Daskalakis and Syrgkanis (2016) where the adversary has only $K$ choices (formally defined in Section 3.1), without further assumptions on the loss function, their FTPL algorithm is *minimax optimal*. To our knowledge this is unknown before.

Second, since $\mathsf{PUCal} = \sup_\ell \mathbb{E}[\mathrm{REG}_\ell] \geq \mathbb{E}[\mathrm{REG}_{\ell'}]$ for any $\ell' \in \mathcal{L}$, Theorem 2 immediately implies a $\Omega(\sqrt{KT})$ lower bound on the pseudo multiclass U-calibration error. In fact, since $\mathsf{UCal} \geq \mathsf{PUCal}$, the same lower bound holds for the actual U-calibration error.

**Corollary 2.** *For any online forecasting algorithm, there exists an oblivious adversary such that* $\mathsf{UCal} \geq \mathsf{PUCal} = \Omega(\sqrt{KT})$.

### 3.3 From PUCal to UCal

We now make an attempt to bound the U-calibration error $\mathsf{UCal}$ of Algorithm 1 for an oblivious adversary. Specifically, since the perturbations are sampled every round and the adversary is oblivious, using Hoeffding's inequality it is straightforward to show that for a fixed $\ell$ and a fixed $\delta \in (0, 1)$, the regret of Algorithm 1 with respect to $\ell$ satisfies $\mathrm{REG}_\ell \leq 4\sqrt{KT} + \sqrt{2T \log{(1/\delta)}}$ with probability at least $1 - \delta$ (see Hutter et al. (2005, Section 9) or Lemma D.1). Therefore, for a finite subset $\mathcal{L}'$ of $\mathcal{L}$, taking a union bound over all $\ell \in \mathcal{L}'$ gives $\sup_{\ell \in \mathcal{L}'} \mathrm{REG}_\ell \leq 4\sqrt{KT} + \sqrt{2T \log{(|\mathcal{L}'|/\delta)}}$ with probability at least $1 - \delta$. Picking $\delta = 1/T$ and using the boundedness of losses, we obtain $\mathsf{UCal}_{\mathcal{L}'} \leq 2 + 4\sqrt{KT} + \sqrt{2T \log{(T |\mathcal{L}'|)}}$. In Appendix D, we generalize this simple argument to any infinite subset $\mathcal{L}'$ of $\mathcal{L}$ with a finite $\epsilon$-covering number $M(\mathcal{L}', \epsilon; \|.\|_\infty)$ and prove for any $\epsilon > 0$,

$$\mathsf{UCal}_{\mathcal{L}'} \leq 2 + 4\epsilon T + 4\sqrt{KT} + \sqrt{2T \log{(T \cdot M(\mathcal{L}', \epsilon; \|.\|_\infty))}}. \tag{2}$$

Using this bound, we now give a concrete example of a simple parameterized family $\mathcal{L}' \subset \mathcal{L}$ for which $\mathsf{UCal}_{\mathcal{L}'} = \mathcal{O}(\sqrt{KT} + \sqrt{T \log T})$. Consider the parameterized class

$$\mathcal{L}' = \{\alpha \ell_1(\boldsymbol{p}, \boldsymbol{y}) + (1 - \alpha)\ell_2(\boldsymbol{p}, \boldsymbol{y}) | \alpha \in [0, 1]\},$$

where $\ell_1(\boldsymbol{p}, \boldsymbol{y}), \ell_2(\boldsymbol{p}, \boldsymbol{y}) \in \mathcal{L}$ are two fixed bounded and proper losses. It is straightforward to verify that $\ell_\alpha(\boldsymbol{p}, \boldsymbol{y}) := \alpha \ell_1(\boldsymbol{p}, \boldsymbol{y}) + (1 - \alpha)\ell_2(\boldsymbol{p}, \boldsymbol{y}) \in \mathcal{L}$, therefore $\mathcal{L}' \subset \mathcal{L}$.

To obtain an $\epsilon \in (0, 1)$ cover for $\mathcal{L}'$, we consider the set $\mathcal{C} := \{0, \epsilon, \ldots, 1 - \epsilon, 1\}$ which partitions the interval $[0, 1]$ to $\frac{1}{\epsilon}$ smaller intervals each of length $\epsilon$. For each $\alpha \in [0, 1]$, let $c_\alpha \in \mathcal{C}$ denote the closest point to $\alpha$ (break ties arbitrarily). Clearly, $|\alpha - c_\alpha| \leq \epsilon$. Next, consider the function $g_\alpha(\boldsymbol{p}, \boldsymbol{y}) := c_\alpha \ell_1(\boldsymbol{p}, \boldsymbol{y}) + (1 - c_\alpha)\ell_2(\boldsymbol{p}, \boldsymbol{y})$. The class $\{g_\alpha(\boldsymbol{p}, \boldsymbol{y}) | \alpha \in [0, 1]\}$ is clearly a $2\epsilon$ cover of $\mathcal{L}'$ with size $\frac{1}{\epsilon}$. Thus, $M(\mathcal{L}', \epsilon; \|.\|_\infty) = \mathcal{O}(\frac{1}{\epsilon})$. It then follows from (2) that

$$\mathsf{UCal}_{\mathcal{L}'} = \mathcal{O}\left(\epsilon T + \sqrt{KT} + \sqrt{T \log\left(\frac{T}{\epsilon}\right)}\right) = \mathcal{O}\left(\sqrt{KT} + \sqrt{T \log T}\right)$$

on choosing $\epsilon = \frac{1}{T}$. On the other hand, in subsection 4.3 we shall argue that for this class with a specific example of $\ell_2$, FTL suffers linear U-calibration error (that is, $\mathsf{UCal}_{\mathcal{L}'} = \Omega(T)$).

## 4 Improved Bounds for Important Sub-Classes

In this section, we show that it is possible to go beyond the $\Theta(\sqrt{KT})$ U-calibration error for several broad sub-classes of $\mathcal{L}$ that include important and common proper losses. These results are achieved by an extremely simple algorithm called Follow-the-Leader (FTL), which at time $t > 1$ forecasts[*]

$$\boldsymbol{p}_t = \frac{1}{t - 1}\sum_{s=1}^{t-1} \boldsymbol{y}_t \in \underset{\boldsymbol{p} \in \Delta_K}{\arg\min} \sum_{s=1}^{t-1} \ell(\boldsymbol{p}, \boldsymbol{y}_s), \tag{3}$$

---

[*]The forecast at time $t = 1$ can be arbitrary.

that is, the average of the past outcomes. For notational convenience, we define $\boldsymbol{n}_t = \sum_{s=1}^{t} \boldsymbol{y}_s$ so that FTL predicts $\boldsymbol{p}_t = \frac{\boldsymbol{n}_{t-1}}{t-1}$, with $\boldsymbol{n}_{t-1,i}$ being the count of outcome $i$ before time $t$.

Importantly, since FTL is a deterministic algorithm, it's PUCal and UCal are always trivially the same. Moreover, there is also no distinction between an oblivious adversary and an adaptive adversary because of this deterministic nature.

### 4.1 Proper Lipschitz Losses

In this section, we show that $\Theta(\log T)$ is the minimax optimal bound for PUCal and UCal for Lipschitz proper losses. Specifically, we consider the following class of $G$-Lipschitz proper losses

$$\mathcal{L}_G \coloneqq \{\ell \in \mathcal{L} \mid |\ell(\boldsymbol{p}, \boldsymbol{y}) - \ell(\boldsymbol{p}', \boldsymbol{y})| \leq G \|\boldsymbol{p} - \boldsymbol{p}'\|, \forall \boldsymbol{p}, \boldsymbol{p}' \in \Delta_K, \boldsymbol{y} \in \mathcal{E}\}.$$

As discussed in Section 2, the two common proper losses, squared loss and spherical loss, are both in $\mathcal{L}_G$ for some $G$. Note that the class of $\mathcal{L}_G$ is rich since according to Lemma 1 it corresponds to the class of concave univariate forms that are Lipschitz and smooth (see Lemma B.2). We now show that FTL enjoys logarithmic U-calibration error with respect to $\mathcal{L}_G$.

**Theorem 3.** *The regret of FTL for learning any $\ell \in \mathcal{L}_G$ is at most $2 + 2G \log T$. Consequently, FTL ensures* $\mathsf{PUCal}_{\mathcal{L}_G} = \mathsf{UCal}_{\mathcal{L}_G} = \mathcal{O}(G \log T)$.

*Proof.* Using the standard Be-the-Leader lemma (see e.g., (Orabona, 2019, Lemma 1.2)) that says $\sum_{t=1}^{T} \ell(\boldsymbol{p}_{t+1}, \boldsymbol{y}_t) \leq \inf_{\boldsymbol{p} \in \Delta_K} \sum_{t=1}^{T} \ell(\boldsymbol{p}, \boldsymbol{y}_t)$, the regret of FTL can be bounded as

$$\mathrm{REG}_\ell \leq 2 + \sum_{t=2}^{T} \ell(\boldsymbol{p}_t, \boldsymbol{y}_t) - \ell(\boldsymbol{p}_{t+1}, \boldsymbol{y}_t) \leq 2 + G \sum_{t=2}^{T} \|\boldsymbol{p}_t - \boldsymbol{p}_{t+1}\|,$$

where the second inequality is because $\ell \in \mathcal{L}_G$. Next, since $\boldsymbol{p}_t = \frac{\boldsymbol{n}_{t-1}}{t-1}$ and $\boldsymbol{p}_{t+1} = \frac{\boldsymbol{n}_t}{t}$, we obtain

$$\mathrm{REG}_\ell \leq 2 + G \sum_{t=2}^{T} \left\| \frac{\boldsymbol{n}_{t-1}}{t-1} - \frac{\boldsymbol{n}_t}{t} \right\| = 2 + G \sum_{t=2}^{T} \left\| \frac{\boldsymbol{n}_{t-1}}{t(t-1)} - \frac{\boldsymbol{y}_t}{t} \right\| \leq 2 + 2G \sum_{t=2}^{T} \frac{1}{t},$$

where the equality follows since $\boldsymbol{n}_t = \boldsymbol{n}_{t-1} + \boldsymbol{y}_t$ and the last inequality follows from the triangle inequality and $\|\boldsymbol{n}_{t-1}\| \leq \|\boldsymbol{n}_{t-1}\|_1 = t - 1$. Finally, since $\sum_{t=2}^{T} \frac{1}{t} \leq \int_1^T \frac{1}{z} dz = \log T$, we obtain $\mathrm{REG}_\ell \leq 2 + 2G \log T$, which completes the proof. □

A closer look at the proof reveals that global Lipschitzness over the entire simplex $\Delta_K$ is in fact not necessary. This is because, for example, in the term $\ell(\boldsymbol{p}_t, \boldsymbol{e}_i) - \ell(\boldsymbol{p}_{t+1}, \boldsymbol{e}_i)$ for some $i \in [K]$, by the definition of FTL the corresponding coordinates $p_{t,i}$ and $p_{t+1,i}$ are almost always at least $1/T$, with only one exception which is when $t$ is the first time we have $\boldsymbol{y}_t = \boldsymbol{e}_i$ and which we can ignore since the regret incurred is at most a constant. This means that having local Lipschitzness in a certain region is enough; see Lemma E.1 for details. Note that the loss induced by the Tsallis entropy (mentioned in Section 2) is exactly one such example where global Lipschitzness does not hold but local Lipschitzness does. We defer the concrete discussion of the regret bounds of FTL on this example to Section 4.2 (where yet another different analysis is introduced).

In the rest of this subsection, we argue that no algorithm can guarantee regret better than $\Omega(\log T)$ for one particular Lipschitz proper loss, making FTL minimax optimal for this class.

**Theorem 4.** *There exists a proper Lipschitz loss $\ell$ such that: for any algorithm* ALG*, there exists a choice of $\boldsymbol{y}_1, \ldots, \boldsymbol{y}_T$ by an oblivious adversary such that the expected regret of* ALG *is $\Omega(\log T)$.*

The loss we use in this lower bound is simply the squared loss $\ell(\boldsymbol{p}, \boldsymbol{y}) = \|\boldsymbol{p} - \boldsymbol{y}\|^2$ with $K = 2$. While squared loss is known to admit $\Theta(\log T)$ regret in other online learning problems such as that from Abernethy et al. (2008), as far as we know there is no study on our setting where the decision set is the simplex and the adversary has only finite choices. It turns out that this variation brings significant technical challenges, and our proof is substantially different from that of Abernethy et al. (2008). We defer the details to Appendix F and discuss the key steps below.

**Step 1:** Since squared loss is convex in $\boldsymbol{p}$, by standard arguments it suffices to consider deterministic algorithms only (see Lemma F.1). Moreover, for deterministic algorithms, there is no difference between an oblivious adversary and an adaptive adversary so that the minimax regret can be written as

$$\text{VAL} = \inf_{\boldsymbol{p}_1 \in \Delta_K} \sup_{\boldsymbol{y}_1 \in \mathcal{E}} \cdots \inf_{\boldsymbol{p}_T \in \Delta_K} \sup_{\boldsymbol{y}_T \in \mathcal{E}} \left[ \sum_{t=1}^{T} \ell(\boldsymbol{p}_t, \boldsymbol{y}_t) - \inf_{\boldsymbol{p} \in \Delta_K} \sum_{t=1}^{T} \ell(\boldsymbol{p}, \boldsymbol{y}_t) \right].$$

To solve this, further define $\mathcal{V}_{\boldsymbol{n},r}$ recursively as $\mathcal{V}_{\boldsymbol{n},r} = \inf_{\boldsymbol{p} \in \Delta_K} \sup_{\boldsymbol{y} \in \mathcal{E}} \mathcal{V}_{\boldsymbol{n}+\boldsymbol{y},r-1} + \ell(\boldsymbol{p}, \boldsymbol{y})$ with $\mathcal{V}_{\boldsymbol{n},0} = -\inf_{\boldsymbol{p} \in \Delta_K} \sum_{i=1}^{K} n_i \ell(\boldsymbol{p}, \boldsymbol{e}_i)$, so that VAL is simply $\mathcal{V}_{\boldsymbol{0},T}$.

**Step 2:** Using the minimax theorem, we further show that $\mathcal{V}_{\boldsymbol{n},r} = \sup_{\boldsymbol{q} \in \Delta_K} \sum_{i=1}^{K} q_i \mathcal{V}_{\boldsymbol{n}+\boldsymbol{e}_i,r-1} + \ell(\boldsymbol{q})$ where the univariate form $\ell(\boldsymbol{q})$ is $1 - \|\boldsymbol{q}\|^2$ (as mentioned in Section 2). Recall that we consider only the binary case $K = 2$, so it is straightforward to give an analytical form of the solution to the maximization over $\boldsymbol{q} \in \Delta_K$. Specifically, writing $\mathcal{V}_{\boldsymbol{n},r} = \mathcal{V}_{(n_1,n_2),r}$ as $\mathcal{V}_{n_1,n_2,r}$ to make notation concise, we show

$$\mathcal{V}_{n_1,n_2,r} = \begin{cases} \mathcal{V}_2 & \text{if } \mathcal{V}_1 - \mathcal{V}_2 < -2, \\ \frac{(\mathcal{V}_1 - \mathcal{V}_2)^2}{8} + \frac{\mathcal{V}_1 + \mathcal{V}_2}{2} + \frac{1}{2} & \text{if } -2 \le \mathcal{V}_1 - \mathcal{V}_2 \le 2, \\ \mathcal{V}_1 & \text{if } \mathcal{V}_1 - \mathcal{V}_2 > 2, \end{cases}$$

where $\mathcal{V}_1$ and $\mathcal{V}_2$ are shorthands for $\mathcal{V}_{n_1+1,n_2,r-1}$ and $\mathcal{V}_{n_1,n_2+1,r-1}$ respectively. Next, by an induction on $r$ we show that for all valid $n_1, n_2, r$ it holds that $-2 \le \mathcal{V}_1 - \mathcal{V}_2 \le 2$ (Lemma F.2), therefore $\mathcal{V}_{n_1,n_2,r}$ is always equal to $\frac{(\mathcal{V}_1 - \mathcal{V}_2)^2}{8} + \frac{\mathcal{V}_1 + \mathcal{V}_2}{2} + \frac{1}{2}$.

**Step 3:** By an induction on $r$ again, we show that $\mathcal{V}_{n_1,n_2,r}$ exhibits a special structure of the form

$$\mathcal{V}_{n_1,n_2,r} = \frac{(n_1 - n_2)^2}{2} \cdot u_r - \frac{2n_1 n_2}{T} + v_r,$$

where $\{u_r\}_{r=0}^{T}$ and $\{v_r\}_{r=0}^{T}$ are recursively defined via $u_{r+1} = u_r + \left( u_r + \frac{1}{T} \right)^2$ and $v_{r+1} = \frac{u_r}{2} + v_r + \frac{r+1}{T} - \frac{1}{2}$ with $u_0 = v_0 = 0$ (Lemma F.3). Since $\text{VAL} = \mathcal{V}_{0,0,T} = v_T$, it remains to show $v_T = \Omega(\log T)$, which is done via two technical lemmas F.4 and F.5.

### 4.2 Decomposable Losses

Next, we consider another sub-class of proper losses that are not necessarily Lipschitz. Instead, their univariate form is decomposable over the $K$ outcomes and additionally satisfies a mild regularity condition. Specifically, we define the following class

$$\mathcal{L}_{dec} := \left\{ \ell \in \mathcal{L} \,\middle|\, \ell(\boldsymbol{p}) \propto \sum_{i=1}^{K} \ell_i(p_i) \text{ where each } \ell_i \text{ is twice continuously differentiable in } (0,1) \right\}.$$

Both the squared loss and its generalization via Tsallis entropy discussed in Section 2 are clearly in this class $\mathcal{L}_{dec}$, with the latter being non-Lipschitz when $\alpha \in (1,2)$. The spherical loss, however, is not decomposable and thus not in $\mathcal{L}_{dec}$. We now show that FTL achieves logarithmic regret against any $\ell \in \mathcal{L}_{dec}$ (see Appendix G for the full proof).

**Theorem 5.** *The regret of FTL for learning any $\ell \in \mathcal{L}_{dec}$ is at most $2K + (K+1)\beta_\ell (1 + \log T)$ for some universal constant $\beta_\ell$ which only depends on $\ell$ and $K$. Consequently, FTL ensures $\mathsf{PUCal}_{\mathcal{L}_{dec}} = \mathsf{UCal}_{\mathcal{L}_{dec}} = \mathcal{O}((\sup_{\ell \in \mathcal{L}_{dec}} \beta_\ell) K \log T)$.*

*Proof Sketch.* We start by showing a certain controlled growth rate of the second derivative of the univariate form (see Appendix H for the proof).

**Lemma 2.** *For a function $f$ that is concave, Lipschitz, and bounded over $[0,1]$ and twice continuously differentiable over $(0,1)$, there exists a constant $c > 0$ such that $|f''(p)| \le c \cdot \max\left( \frac{1}{p}, \frac{1}{1-p} \right)$ for all $p \in (0,1)$.*

Note that according to Lemma 1, each $\ell_i$ must be concave, Lipschitz, and bounded, for the induced loss to be proper and bounded. Therefore, using Lemma 2, there exists a constant $c_i > 0$ such that $|\ell_i''(p)| \leq c_i \max\left(\frac{1}{p}, \frac{1}{1-p}\right)$ for each $i$. The rest of the proof in fact only relies on this property; in other words, the regret bound holds even if one replaces the twice continuous differentiability condition with this (weaker) property.

More specifically, for each $i \in [K]$, let $\mathcal{T}_i := \{t_{i,1}, \ldots, t_{i,k_i}\} \subset [T]$ be the subset of rounds where the true outcome is $i$ (which could be empty). Then, using the Be-the-Leader lemma again and trivially bounding the regret by its maximum value for the (at most $K$) rounds when an outcome appears for the first time, we obtain

$$\text{REG}_\ell \leq 2K + \sum_{i=1}^{K} \sum_{t \in \mathcal{T}_i \setminus \{t_{i,1}\}} \underbrace{\ell(\boldsymbol{p}_t, \boldsymbol{e}_i) - \ell(\boldsymbol{p}_{t+1}, \boldsymbol{e}_i)}_{\delta_{t,i}}.$$

By using the characterization result in Lemma 1, we then express $\ell(\boldsymbol{p}_t, \boldsymbol{e}_i)$ and $\ell(\boldsymbol{p}_{t+1}, \boldsymbol{e}_i)$ in terms of the univariate forms $\ell(\boldsymbol{p}_t), \ell(\boldsymbol{p}_{t+1})$, and their respective gradients $\nabla\ell(\boldsymbol{p}_t), \nabla\ell(\boldsymbol{p}_{t+1})$. Next, using the concavity of $\ell_i$, the Mean Value Theorem, and Lemma 2, we argue that

$$\delta_{t,i} \leq \sum_{j=1}^{K} \beta_{\ell,j} \cdot |p_{t+1,j} - p_{t,j}| \cdot \max\left(\frac{1}{\xi_{t,j}}, \frac{1}{1 - \xi_{t,j}}\right), \tag{4}$$

for some $\boldsymbol{\xi}_t$ that is a convex combination of $\boldsymbol{p}_t$ and $\boldsymbol{p}_{t+1}$, and constant $\beta_{\ell,i} = \tilde{c}_K \cdot c_i$ ($\tilde{c}_K$ is the scaling constant such that $\ell(\boldsymbol{p}) = \tilde{c}_K \sum_{i=1}^{K} \ell_i(p_i)$). To bound (4), we consider the terms $\frac{|p_{t+1,j} - p_{t,j}|}{\xi_{t,j}}$ and $\frac{|p_{t+1,j} - p_{t,j}|}{1 - \xi_{t,j}}$ individually and find that they are always bounded by either $\frac{1}{n_{t-1,i}}$ or $\frac{1}{t-1}$ according to the update rule of FTL. Thus, we obtain

$$\text{REG}_\ell \leq 2K + \beta_\ell(\mathcal{S}_1 + \mathcal{S}_2), \text{ where } \mathcal{S}_1 = \sum_{i=1}^{K} \sum_{t \in \mathcal{T}_i \setminus \{t_{i,1}\}} \frac{1}{n_{t-1,i}}, \mathcal{S}_2 = \sum_{i=1}^{K} \sum_{t \in \mathcal{T}_i \setminus \{t_{i,1}\}} \frac{1}{t-1},$$

and $\beta_\ell = \sum_{i=1}^{K} \beta_{\ell,i}$. Finally, direct calculation shows $\mathcal{S}_1 \leq K(1 + \log T)$ and $\mathcal{S}_2 \leq 1 + \log T$, which finishes the proof. $\qquad\square$

To showcase the usefulness of this result, we go back to the Tsallis entropy example.

**Corollary 3.** *For any loss $\ell$ with univariate form $\ell(\boldsymbol{p}) = -\tilde{c}_K \sum_{i=1}^{K} p_i^\alpha$ for $\alpha \in (1, 2)$ (the constant $\tilde{c}_K$ is such that the loss has range $[-1, 1]$), FTL ensures $\text{REG}_\ell = \mathcal{O}(\tilde{c}_K \alpha(\alpha - 1)K^2 \log T)$.*

*Proof.* As mentioned, our proof of Theorem 5 only relies on Lemma 2, and it is straightforward to verify that for the loss considered here, one can take the constant $c$ in Lemma 2 to be $\alpha(\alpha - 1)$, and thus the regret of FTL is $\mathcal{O}(K\beta_\ell \log T)$ with $\beta_\ell = K\tilde{c}_K\alpha(\alpha - 1)$. $\qquad\square$

On the other hand, if one were to use the proof based on local Lipschitzness (mentioned in Section 4.1 and discussed in Appendix E), one would only obtain a regret bound of order $\mathcal{O}(K + \tilde{c}_K\alpha(\alpha - 1)T^{2-\alpha} \log T)$, which is much worse (especially for small $\alpha$). Finally, we remark that for $\alpha \geq 2$, the bivariate form is Lipschitz, and thus FTL also ensures logarithmic regret according to Theorem 3.

### 4.3 FTL Cannot Handle General Proper Losses

Despite yielding improved regret for Lipschitz and other special classes of proper losses, unfortunately, FTL is not a good algorithm in general when dealing with proper losses, as shown below.

**Theorem 6.** *There exists a proper loss $\ell$ and a choice of $\boldsymbol{y}_1, \ldots, \boldsymbol{y}_T$ by an oblivious adversary such that the regret $\text{REG}_\ell$ of FTL is $\Omega(T)$.*

*Proof.* The loss we consider is in fact the same V-shaped loss used in the proof of Theorem 2 that shows all algorithms must suffer $\Omega(\sqrt{KT})$ regret. Here, we show that FTL even suffers linear regret

for this loss. Specifically, it suffices to consider the binary case $K = 2$ and the univariate form $\ell(\boldsymbol{p}) = -\frac{1}{2}\left(\left|p_1 - \frac{1}{2}\right| + \left|p_2 - \frac{1}{2}\right|\right)$. Using Lemma 1, we obtain the following bivariate form:

$$\ell(\boldsymbol{p}, \boldsymbol{e}_1) = -\frac{1}{2}\mathrm{sign}\left(p_1 - \frac{1}{2}\right), \quad \ell(\boldsymbol{p}, \boldsymbol{e}_2) = -\frac{1}{2}\mathrm{sign}\left(p_2 - \frac{1}{2}\right),$$

where the sign function is defined as $\mathrm{sign}(x) = 1$ if $x > 0$; $-1$ if $x < 0$; $0$ if $x = 0$. Therefore, $\ell(\boldsymbol{p}, \boldsymbol{e}_1)$ is equal to $\frac{1}{2}$ if $p_1 < \frac{1}{2}$; $0$ if $p_1 = \frac{1}{2}$; $-\frac{1}{2}$ if $p_1 \geq \frac{1}{2}$. Similarly, $\ell(\boldsymbol{p}, \boldsymbol{e}_2)$ is equal to $-\frac{1}{2}$ if $p_1 \leq \frac{1}{2}$; $0$ if $p_1 = \frac{1}{2}$; $\frac{1}{2}$ if $p_1 \geq \frac{1}{2}$. Let $T$ be even and $\boldsymbol{y}_t = \boldsymbol{e}_1$ if $t$ is odd, and $\boldsymbol{e}_2$ otherwise. For such a sequence $\boldsymbol{y}_1, \ldots, \boldsymbol{y}_T$, the benchmark selects $\boldsymbol{\beta} = \frac{1}{T}\sum_{t=1}^T \boldsymbol{y}_t = [\frac{1}{2}, \frac{1}{2}]$ and incurs $0$ cost. On the other hand, FTL chooses $\boldsymbol{p}_t = [\frac{1}{2}, \frac{1}{2}]$ when $t$ is odd, and $\boldsymbol{p}_t = \left[\frac{t}{2(t-1)}, \frac{t-2}{2(t-1)}\right]$ otherwise. Thus, the regret of FTL is $\mathrm{REG}_\ell = \sum_{t=2}^T \ell(\boldsymbol{p}_t, \boldsymbol{e}_2) = \frac{T}{4}$. This completes the proof. □

Consider the parametrized class in subsection 3.3, let $K = 2$, $\ell_2(\boldsymbol{p}, \boldsymbol{y})$ correspond to the V-shaped loss in Theorem 6, and consider any $\ell_1(\boldsymbol{p}, \boldsymbol{y}) \in \mathcal{L}$. It follows from Theorem 6 that $\mathrm{REG}_{\ell_\alpha} = \Omega(T)$ when $\alpha = 0$, therefore $\mathsf{UCal}_{\mathcal{L}'} = \sup_{\alpha \in [0,1]} \mathrm{REG}_{\ell_\alpha} = \Omega(T)$, whereas Algorithm 1 ensures $\mathsf{UCal}_{\mathcal{L}'} = \mathcal{O}(\sqrt{T \log T})$.

# 5 Conclusion and Future Directions

In this paper, we give complete answers to various questions regarding the minimax optimal bounds on multiclass U-calibration error, a notion of simultaneous loss minimization proposed by (Kleinberg et al., 2023) for the fundamental problem of making online forecasts on unknown outcomes. We not only improve their $\mathsf{PUCal} = \mathcal{O}(K\sqrt{T})$ upper bound and show that the minimax pseudo U-calibration error is $\Theta(\sqrt{KT})$, but also further show that logarithmic U-calibration error can be achieved by an extremely simple algorithm for several important classes of proper losses.

There are many interesting future directions, including 1) understanding the optimal bound on the actual U-calibration error $\mathsf{UCal}$, 2) generalizing the results to losses that are not necessarily proper, and 3) studying the contextual case and developing more efficient algorithms with better bounds compared to those in the recent work of Garg et al. (2024).

## Acknowledgements

We thank mathoverflow user mathworker21 for their help in formulating and proving Lemma F.5. Haipeng Luo was supported by NSF award IIS-1943607 and a Google Research Scholar Award, and Vatsal Sharan was supported by NSF CAREER Award CCF-2239265 and an Amazon Research Award. Any opinions, findings, and conclusions or recommendations expressed in this material are those of the author(s) and do not reflect the views of Amazon.

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

# A  Concentration and Anti-Concentration Inequalities

**Lemma A.1** (Markov's inequality). *For a non-negative random variable $X$, and any $a > 0$, we have* $\mathbb{P}(X \geq a) \leq \frac{\mathbb{E}[X]}{a}$.

**Lemma A.2** (Khintchine's inequality). *Let $\epsilon_1, \ldots, \epsilon_T$ be i.i.d. Rademacher random variables, i.e., $\mathbb{P}(\epsilon_i = +1) = \mathbb{P}(\epsilon_i = -1) = \frac{1}{2}$ for all $i \in [T]$. Then,*

$$\mathbb{E}\left|\sum_{i=1}^{T} \epsilon_i x_i\right| \geq \frac{1}{\sqrt{2}}\left(\sum_{i=1}^{T} x_i^2\right)^{\frac{1}{2}}$$

*for any $x_1, \ldots, x_T \in \mathbb{R}$.*

**Lemma A.3** (Hoeffding's inequality). *Let $X_1, \ldots, X_T$ be independent random variables satisfying $a_i \leq X_i \leq b_i$ for all $i \in [T]$. Then, for any $\epsilon > 0$, we have*

$$\mathbb{P}\left(\sum_{t=1}^{T} X_T - \mathbb{E}[X_t] \geq \epsilon\right) \leq \exp\left(-\frac{2\epsilon^2}{\sum_{i=1}^{T}(b_i - a_i)^2}\right), and$$

$$\mathbb{P}\left(\sum_{t=1}^{T} X_T - \mathbb{E}[X_t] \leq -\epsilon\right) \leq \exp\left(-\frac{2\epsilon^2}{\sum_{i=1}^{T}(b_i - a_i)^2}\right).$$

**Lemma A.4** (Reverse Chernoff bounds). *([Klein and Young, 1999](), Lemma 5.2) Let $\bar{X} := \frac{1}{T}\sum_{i=1}^{T} X_i$ be the average of $T$ i.i.d. Bernoulli random variables $X_1, \ldots, X_T$ with $\mathbb{E}[X_i] = p$ for all $i \in [T]$. If $\epsilon \in (0, \frac{1}{2}], p \in (0, \frac{1}{2}]$ are such that $\epsilon^2 pT \geq 3$, then*

$$\mathbb{P}(\bar{X} \leq (1 - \epsilon)p) \geq \exp(-9\epsilon^2 pT), \quad \mathbb{P}(\bar{X} \geq (1 + \epsilon)p) \geq \exp(-9\epsilon^2 pT).$$

# B  Deferred Proofs for Proper Losses

**Lemma B.1.** *For a proper loss $\ell(\boldsymbol{p}, \boldsymbol{y})$, the following holds true for any $n \in \mathbb{N}$ and $\boldsymbol{y}_1, \ldots, \boldsymbol{y}_n \in \mathcal{E}$:*

$$\frac{1}{n}\sum_{i=1}^{n} \boldsymbol{y}_i \in \underset{\boldsymbol{p} \in \Delta_K}{\operatorname{argmin}} \frac{1}{n}\sum_{i=1}^{n} \ell(\boldsymbol{p}, \boldsymbol{y}).$$

*Proof.* Let $\boldsymbol{\beta} := \frac{1}{n}\sum_{i=1}^{n} \boldsymbol{y}_i$. Since $\ell$ is proper, $\mathbb{E}_{\boldsymbol{y} \sim \boldsymbol{\beta}}[\ell(\boldsymbol{\beta}, \boldsymbol{y})] \leq \mathbb{E}_{\boldsymbol{y} \sim \boldsymbol{\beta}}[\ell(\boldsymbol{p}', \boldsymbol{y})]$ for all $\boldsymbol{p}' \in \Delta_K$. Notably, for any $\boldsymbol{p} \in \Delta_K$, we have

$$\sum_{i=1}^{K} \beta_i \ell(\boldsymbol{p}, \boldsymbol{e}_i) = \frac{1}{n}\sum_{i=1}^{K}\sum_{j=1}^{n} \mathbf{1}[\boldsymbol{y}_j = \boldsymbol{e}_i]\ell(\boldsymbol{p}, \boldsymbol{e}_i) = \frac{1}{n}\sum_{j=1}^{n}\sum_{i=1}^{K} \mathbf{1}[\boldsymbol{y}_j = \boldsymbol{e}_i]\ell(\boldsymbol{p}, \boldsymbol{e}_i) = \frac{1}{n}\sum_{j=1}^{n} \ell(\boldsymbol{p}, \boldsymbol{y}_j).$$

Thus, $\frac{1}{n}\sum_{j=1}^{n} \ell(\boldsymbol{\beta}, \boldsymbol{y}_j) \leq \frac{1}{n}\sum_{j=1}^{n} \ell(\boldsymbol{p}', \boldsymbol{y}_j)$ for any $\boldsymbol{p}' \in \Delta_K$. This completes the proof. $\square$

**Lemma B.2.** *Let $\ell(\boldsymbol{p})$ be a differentiable and concave, $\alpha$-Lipschitz function over $\Delta_K$ such that $\nabla \ell(\boldsymbol{p})$ is $\beta$-Lipschitz over $\Delta_K$. Then, $\ell(\boldsymbol{p}, \boldsymbol{y})$ is proper and $(2\alpha + 2\beta)$-Lipschitz in $\boldsymbol{p} \in \Delta_K$ for each $\boldsymbol{y} \in \mathcal{E}$.*

*Proof.* Since $\ell(\boldsymbol{p})$ is concave, it follows from Lemma 1 that $\ell(\boldsymbol{p}, \boldsymbol{y})$ is proper. To prove the second part, without any loss of generality, assume that $\boldsymbol{y} = \boldsymbol{e}_1$. For any $\boldsymbol{p}, \boldsymbol{p}' \in \Delta_K$, we have

$$\begin{aligned}
\ell(\boldsymbol{p}, \boldsymbol{e}_1) - \ell(\boldsymbol{p}', \boldsymbol{e}_1) &= \ell(\boldsymbol{p}) + \langle\nabla\ell(\boldsymbol{p}), \boldsymbol{e}_1 - \boldsymbol{p}\rangle - (\ell(\boldsymbol{p}') + \langle\nabla\ell(\boldsymbol{p}'), \boldsymbol{e}_1 - \boldsymbol{p}'\rangle) \\
&\leq \alpha\|\boldsymbol{p} - \boldsymbol{p}'\| + \nabla_1\ell(\boldsymbol{p}) - \nabla_1\ell(\boldsymbol{p}') - (\langle\boldsymbol{p}, \nabla\ell(\boldsymbol{p})\rangle - \langle\boldsymbol{p}', \nabla\ell(\boldsymbol{p}')\rangle) \\
&\leq (\alpha + \beta)\|\boldsymbol{p} - \boldsymbol{p}'\| + \langle\boldsymbol{p}' - \boldsymbol{p}, \nabla\ell(\boldsymbol{p}')\rangle + \langle\boldsymbol{p}, \nabla\ell(\boldsymbol{p}') - \nabla\ell(\boldsymbol{p})\rangle \\
&\leq (\alpha + \beta)\|\boldsymbol{p} - \boldsymbol{p}'\| + \|\boldsymbol{p}' - \boldsymbol{p}\|\|\nabla\ell(\boldsymbol{p}')\| + \|\boldsymbol{p}\|\|\nabla\ell(\boldsymbol{p}') - \nabla\ell(\boldsymbol{p})\| \\
&\leq 2(\alpha + \beta)\|\boldsymbol{p} - \boldsymbol{p}'\|,
\end{aligned}$$

where the first equality follows from Lemma 1; the first inequality follows from the Lipschitzness of $\ell$; the second inequality follows from the Lipschitzness of $\nabla\ell(\boldsymbol{p})$; the third inequality follows from the Cauchy-Schwartz inequality; the final inequality follows since $\|\boldsymbol{p}\| \le 1$ and $\nabla\ell(\boldsymbol{p})$ is Lipschitz. This completes the proof. $\square$

**Proposition B.1.** *The spherical loss is $\sqrt{K}$-Lipschitz.*

*Proof.* We shall show that $\|\nabla\ell(\boldsymbol{p},\boldsymbol{y})\| \le \sqrt{K}$ for all $\boldsymbol{y} \in \mathcal{E}, \boldsymbol{p} \in \Delta_K$, where the gradient is taken with respect to the first argument. Without any loss of generality, assume $\boldsymbol{y} = \boldsymbol{e}_1$, thus $\ell(\boldsymbol{p},\boldsymbol{y}) = -\frac{p_1}{\|\boldsymbol{p}\|}$. It is easy to obtain the following:

$$\frac{\partial\ell(\boldsymbol{p},\boldsymbol{y})}{\partial p_1} = -\frac{\|\boldsymbol{p}\|^2 - p_1^2}{\|\boldsymbol{p}\|^3}, \quad \frac{\partial\ell(\boldsymbol{p},\boldsymbol{y})}{\partial p_i} = \frac{p_1 p_i}{\|\boldsymbol{p}\|^3} \quad \text{for all } i > 2.$$

Thus, $\|\nabla\ell(\boldsymbol{p},\boldsymbol{e}_1)\| = \frac{1}{\|\boldsymbol{p}\|^3}\sqrt{(\|\boldsymbol{p}\|^2 - p_1^2)^2 + p_1^2\sum_{i=2}^K p_i^2} = \frac{1}{\|\boldsymbol{p}\|^2}\sqrt{(\|\boldsymbol{p}\|^2 - p_1^2)} \le \frac{1}{\|\boldsymbol{p}\|} \le \sqrt{K}$, where the last inequality follows the Cauchy-Schwartz inequality. $\square$

## C   Proof of Theorem 2

**Theorem 2.** *There exists a proper loss $\ell$ with range $[-1, 1]$ such that the following holds: for any online algorithm $\mathsf{ALG}$, there exists a choice of $\boldsymbol{y}_1, \ldots, \boldsymbol{y}_T$ by an oblivious adversary such that the expected regret $\mathbb{E}[\mathrm{REG}_\ell]$ of $\mathsf{ALG}$ is $\Omega(\sqrt{KT})$ when $T \ge 12K$.*

*Proof.* Consider the function defined as

$$\ell(\boldsymbol{p}) := -\frac{1}{2}\sum_{i=1}^K \left| p_i - \frac{1}{K} \right|.$$

It follows from Lemma 1 that the bivariate form of $\ell$ is $\ell(\boldsymbol{p},\boldsymbol{y}) = \ell(\boldsymbol{p}) + \langle \boldsymbol{g_p}, \boldsymbol{p} - \boldsymbol{y} \rangle$, where $\boldsymbol{g_p}$ denotes a subgradient of $\ell(\boldsymbol{p})$ at $\boldsymbol{p}$. For the choice of $\ell$, $g_i = -\frac{1}{2}\mathrm{sign}(p_i - \frac{1}{K})$, where the sign function is defined as $\mathrm{sign}(x) = 1$ if $x > 0$; $-1$ if $x < 0$; $0$ if $x = 0$. We first show that $\ell(\boldsymbol{p},\boldsymbol{y}) \in [-1, 1]$. Without any loss of generality, assume $\boldsymbol{y} = \boldsymbol{e}_1$. Therefore,

$$\ell(\boldsymbol{p},\boldsymbol{e}_1) = \frac{1}{2}\left[ -\sum_{i=1}^K \left| p_i - \frac{1}{K} \right| - (1 - p_1)\mathrm{sign}\left( p_1 - \frac{1}{K} \right) + \sum_{i=2}^K p_i\,\mathrm{sign}\left( p_i - \frac{1}{K} \right) \right]$$

$$= \frac{1}{2}\left[ -\left| p_1 - \frac{1}{K} \right| - (1 - p_1)\mathrm{sign}\left( p_1 - \frac{1}{K} \right) + \sum_{i=2}^K p_i\,\mathrm{sign}\left( p_i - \frac{1}{K} \right) - \left| p_i - \frac{1}{K} \right| \right]$$

$$= \frac{1}{2}\left[ -\left( 1 - \frac{1}{K} \right)\mathrm{sign}\left( p_1 - \frac{1}{K} \right) + \frac{1}{K}\sum_{i=2}^K \mathrm{sign}\left( p_i - \frac{1}{K} \right) \right].$$

It is then trivial to note that $\boldsymbol{p} = \boldsymbol{e}_1$ corresponds to a minimum, with value $\ell(\boldsymbol{e}_1,\boldsymbol{e}_1) = -\frac{K-1}{K}$; $\boldsymbol{p} = [0, \frac{1}{K-1}, \ldots, \frac{1}{K-1}]$ corresponds to a maximum, with value $\frac{K-1}{K}$.

Next, we consider a randomized oblivious adversary which samples $\boldsymbol{y}_1, \ldots, \boldsymbol{y}_T$ from the uniform distribution over $\mathcal{E}$. For such an adversary, we shall show that $\mathbb{E}[\mathrm{REG}] = \Omega(\sqrt{KT})$. We overload the notation and use $\mathsf{ALG}_{1:t}$ to denote the internal randomness of the algorithm until time $t$ (inclusive). In particular, this notation succintly represents both deterministic and randomized algorithms ($\boldsymbol{p}_t$ could be sampled from a distribution $\mathcal{D}_t$). Similarly, $\mathsf{ALG}_t$ shall denote the randomness at time $t$.

With this notation, in the constructed randomized environment, the expected cost of ALG is

$$\mathbb{E}\left[\sum_{t=1}^{T}\ell(\boldsymbol{p}_t,\boldsymbol{y}_t)\right] = \mathbb{E}_{\mathsf{ALG}_{1:T},\boldsymbol{y}_1,\dots,\boldsymbol{y}_T}\left[\sum_{t=1}^{T}\ell(\boldsymbol{p}_t,\boldsymbol{y}_t)\right]$$

$$= \sum_{t=1}^{T}\mathbb{E}_{\mathsf{ALG}_{1:T},\boldsymbol{y}_1,\dots,\boldsymbol{y}_T}[\ell(\boldsymbol{p}_t,\boldsymbol{y}_t)]$$

$$= \sum_{t=1}^{T}\mathbb{E}_{\mathsf{ALG}_{1:t-1},\boldsymbol{y}_1,\dots,\boldsymbol{y}_{t-1}}\mathbb{E}_{\mathsf{ALG}_t,\boldsymbol{y}_t}[\ell(\boldsymbol{p}_t,\boldsymbol{y}_t)|\mathsf{ALG}_{1:t-1},\boldsymbol{y}_1,\dots,\boldsymbol{y}_{t-1}]$$

$$= \sum_{t=1}^{T}\mathbb{E}_{\mathsf{ALG}_{1:t-1},\boldsymbol{y}_1,\dots,\boldsymbol{y}_{t-1}}\mathbb{E}_{\mathsf{ALG}_t}\mathbb{E}_{\boldsymbol{y}_t}[\ell(\boldsymbol{p}_t,\boldsymbol{y}_t)|\mathsf{ALG}_{1:t-1},\boldsymbol{y}_1,\dots,\boldsymbol{y}_{t-1}]$$

$$= \sum_{t=1}^{T}\mathbb{E}_{\mathsf{ALG}_{1:t-1},\boldsymbol{y}_1,\dots,\boldsymbol{y}_{t-1}}\mathbb{E}_{\mathsf{ALG}_t}\left[\frac{1}{K}\sum_{i=1}^{K}\ell(\boldsymbol{p}_t,\boldsymbol{e}_i)\right] \geq 0,$$

where in the first equality we have made the randomness explicit; the second equality follows from the linearity of expectations; the third equality follows from the law of iterated expectations; the fourth equality follows because $\mathsf{ALG}_t$ is independent of $\boldsymbol{y}_t$ ($\boldsymbol{p}_t$ is chosen without knowing $\boldsymbol{y}_t$) and vice-versa ($\boldsymbol{y}_t$ is sampled uniformly randomly from $\mathcal{E}$); the fifth equality follows by expanding out the expectation; the first inequality follows since $\sum_{i=1}^{K}\ell(\boldsymbol{p},\boldsymbol{e}_i) \geq 0$ for any $\boldsymbol{p} \in \Delta_K$. This is because,

$$\sum_{i=1}^{K}\ell(\boldsymbol{p},\boldsymbol{e}_i) = K\ell(\boldsymbol{p}) + \sum_{i=1}^{K}\langle\boldsymbol{g_p},\boldsymbol{e}_i-\boldsymbol{p}\rangle = K\left(\ell(\boldsymbol{p}) + \left\langle\boldsymbol{g_p},\frac{1}{K}\cdot\mathbf{1}_K-\boldsymbol{p}\right\rangle\right),$$

where $\mathbf{1}_K$ denotes the $K$-dimensional vector of all ones. Next, since $\ell(\boldsymbol{p})$ is concave over $\Delta_K$, the term above can be lower bounded by $K\ell\left(\frac{1}{K}\cdot\mathbf{1}_K\right) = 0$.

Next, the expected regret of ALG can be lower bounded in the following manner:

$$\mathbb{E}[\mathrm{REG}_\ell] = \mathbb{E}_{\mathsf{ALG}_{1:T},\boldsymbol{y}_1,\dots,\boldsymbol{y}_T}\left[\sum_{t=1}^{T}\ell(\boldsymbol{p}_t,\boldsymbol{y}_t) - \inf_{\boldsymbol{p}\in\Delta_K}\sum_{t=1}^{T}\ell(\boldsymbol{p},\boldsymbol{y}_t)\right]$$

$$\geq -\mathbb{E}_{\boldsymbol{y}_1,\dots,\boldsymbol{y}_T}\left[\inf_{\boldsymbol{p}\in\Delta_K}\sum_{t=1}^{T}\ell(\boldsymbol{p},\boldsymbol{y}_t)\right]$$

$$= -\mathbb{E}_{\boldsymbol{y}_1,\dots,\boldsymbol{y}_T}\left[\sum_{t=1}^{T}\ell\left(\frac{1}{T}\sum_{t=1}^{T}\boldsymbol{y}_t,\boldsymbol{y}_t\right)\right], \tag{5}$$

where the first inequality follows since the expected cost of ALG is non-negative, and the benchmark is independent of ALG; the second equality follows from property 1. In the next steps, we deal with the expectation in (5). Sample $\boldsymbol{y}_1,\dots,\boldsymbol{y}_T$ from $\mathcal{E}$ and let $n_1,\dots,n_K$ denote the counts of the $K$ basis vectors, i.e., $n_i = |\{j \in [T]; \boldsymbol{y}_j = \boldsymbol{e}_i\}|$. Clearly, $\sum_{i=1}^{K}n_i = T$. Let $\boldsymbol{n} = [n_1,\dots,n_K]$ collect these counts. Then, $\frac{1}{T}\sum_{t=1}^{T}\boldsymbol{y}_t = [\frac{n_1}{T},\dots,\frac{n_K}{T}] = \frac{1}{T}\cdot\boldsymbol{n}$, and

$$\sum_{t=1}^{T}\ell\left(\frac{1}{T}\sum_{t=1}^{T}\boldsymbol{y}_t,\boldsymbol{y}_t\right) = \sum_{i=1}^{K}n_i\ell\left(\frac{1}{T}\cdot\boldsymbol{n},\boldsymbol{e}_i\right)$$

$$= \sum_{i=1}^{K}n_i\left(\ell\left(\frac{1}{T}\cdot\boldsymbol{n}\right) + \left\langle\boldsymbol{g}_{\frac{1}{T}\cdot\boldsymbol{n}},\boldsymbol{e}_i-\frac{1}{T}\cdot\boldsymbol{n}\right\rangle\right),$$

$$= T\ell\left(\frac{1}{T}\cdot\boldsymbol{n}\right) = -\frac{1}{2}\sum_{i=1}^{K}\left|n_i-\frac{T}{K}\right|.$$

Thus, the term in (5) equals $\frac{1}{2} \cdot \mathbb{E}_{n_1,\ldots,n_K} \left[ \sum_{i=1}^{K} \left| n_i - \frac{T}{K} \right| \right]$ where $n_1, \ldots, n_K$ are sampled from a multinomial distribution with event probability equal to $\frac{1}{K}$. Further, using the linearity of expectations we arrive at

$$\mathbb{E}[\text{REG}_\ell] \geq \frac{1}{2} \sum_{i=1}^{K} \mathbb{E}_{n_i} \left| n_i - \frac{T}{K} \right| = \frac{K}{2} \cdot \mathbb{E} \left[ \left| \sum_{t=1}^{T} X_t - \mathbb{E}[X_t] \right| \right],$$

where $X_t$ is a Bernoulli random variable with mean $\frac{1}{K}$. Next, we bound $\mathbb{E} \left[ \left| \sum_{t=1}^{T} X_t - \mathbb{E}[X_t] \right| \right]$ using an anti-concentration bound on Bernoulli random variables. In particular, applying Lemma A.4, we obtain

$$\mathbb{P} \left( \left| \sum_{t=1}^{T} X_t - \mathbb{E}[X_t] \right| \geq a \right) \geq 2 \exp \left( -\frac{9a^2 K}{T} \right)$$

for any $a \in \left[ \sqrt{\frac{3T}{K}}, \frac{T}{2K} \right]$. When $T \geq 12K$, this interval is non-empty. From the Markov's inequality (Lemma A.1), we have

$$\mathbb{E} \left[ \left| \sum_{t=1}^{T} X_t - \mathbb{E}[X_t] \right| \right] \geq a \mathbb{P} \left( \left| \sum_{t=1}^{T} X_t - \mathbb{E}[X_t] \right| \geq a \right)$$

for any $a > 0$. Setting $a = \sqrt{\frac{3T}{K}}$ we arrive at $\mathbb{E} \left[ \left| \sum_{t=1}^{T} X_t - \mathbb{E}[X_t] \right| \right] \geq 2\sqrt{3} \exp(-27) \sqrt{\frac{T}{K}}$. Thus,

$$\mathbb{E}[\text{REG}_\ell] \geq \sqrt{3} \exp(-27) \sqrt{KT},$$

which completes the proof. □

**Remark C.1.** *For $K = 2$, the use of Lemma A.4 can be sidestepped via the use of Khintchine's inequality. Indeed, in this case we have*

$$\mathbb{E}[\text{REG}_\ell] \geq \mathbb{E}_n \left| n - \frac{T}{2} \right| = \frac{1}{2} \cdot \mathbb{E}_{\epsilon_1,\ldots,\epsilon_T} \left| \sum_{t=1}^{T} \epsilon_t \right| \geq \sqrt{\frac{T}{8}},$$

*where $\epsilon_1, \ldots, \epsilon_T$ are i.i.d. Rademacher random variables, and the last inequality follows from Khintchine's inequality (Lemma A.2).*

## D   Bounding the (Actual) Multiclass U-Calibration Error

In this section, we bound the U-calibration error $\text{UCal}_{\mathcal{L}'}$ for subclasses $\mathcal{L}'$ of $\mathcal{L}$ with a finite covering number. We begin with deriving a high probability bound on the regret of Algorithm 1.

**Lemma D.1.** *Fix some $\ell \in \mathcal{L}$. Then, for any $\delta \in (0, 1)$, the regret of Algorithm 1 satisfies*

$$\text{REG}_\ell \leq 4\sqrt{KT} + \sqrt{2T \log \left( \frac{1}{\delta} \right)},$$

*with probability at least $1 - \delta$.*

*Proof.* Define the random variable $X_t := \ell(\boldsymbol{p}_t, \boldsymbol{y}_t)$, thus $X_t \in [-1, 1]$. Since the adversary is oblivious and $m_{t,1}, \ldots, m_{t,K}$ are sampled every round independently, $X_1, \ldots, X_T$ are independent. Applying Hoeffdings inequality (Lemma A.3), for any $\epsilon > 0$, we have

$$\mathbb{P} \left( \sum_{t=1}^{T} \ell(\boldsymbol{p}_t, \boldsymbol{y}_t) - \mathbb{E}[\ell(\boldsymbol{p}_t, \boldsymbol{y}_t)] \geq \epsilon \right) \leq \exp \left( -\frac{\epsilon^2}{2T} \right),$$

which implies that $\mathbb{P} (\text{REG}_\ell - \mathbb{E}[\text{REG}_\ell] \leq \epsilon) \geq 1 - \exp \left( -\frac{\epsilon^2}{2T} \right)$. Let $\delta := \exp \left( -\frac{\epsilon^2}{2T} \right)$. Then,

$$\mathbb{P} \left( \text{REG}_\ell - \mathbb{E}[\text{REG}_\ell] \leq \sqrt{2T \log \left( \frac{1}{\delta} \right)} \right) \geq 1 - \delta.$$

Finally, applying the result of Theorem 1 to bound $\mathbb{E}[\text{REG}_\ell]$ completes the proof. □

Using this high probability bound, we first bound $\mathsf{UCal}_{\mathcal{L}'}$ when $\mathcal{L}'$ is a finite subset of $\mathcal{L}$.

**Lemma D.2.** *Fix a subset $\mathcal{L}' \subset \mathcal{L}$ with $|\mathcal{L}'| < \infty$. Algorithm 1 ensures*

$$\mathsf{UCal}_{\mathcal{L}'} \leq 2 + 4\sqrt{KT} + \sqrt{2T \log\left(T |\mathcal{L}'|\right)}.$$

*Proof.* For any $\ell \in \mathcal{L}$, let $\mathcal{E}_\ell$ denote the event that $\mathrm{REG}_\ell \leq 4\sqrt{KT} + \sqrt{2T \log\left(\frac{1}{\delta}\right)}$. From Lemma D.1, $\mathbb{P}\left(\mathcal{E}_\ell\right) \geq 1 - \delta$. Let $\mathcal{S} := \sup_{\ell \in \mathcal{L}'} \mathrm{REG}_\ell$. Thus, the probability that $\mathcal{S}$ is bounded by the same quantity is $\mathbb{P}\left(\cap_{\ell \in \mathcal{L}'} \mathcal{E}_\ell\right)$, which can be bounded as

$$\mathbb{P}\left(\cap_{\ell \in \mathcal{L}'} \mathcal{E}_\ell\right) = 1 - \mathbb{P}\left(\cup_{\ell \in \mathcal{L}'} \mathcal{E}'_\ell\right) \geq 1 - \sum_{\ell \in \mathcal{L}'} \mathbb{P}\left(\mathcal{E}'_\ell\right) = \sum_{\ell \in \mathcal{L}'} \mathbb{P}\left(\mathcal{E}_\ell\right) + 1 - |\mathcal{L}'| \geq 1 - |\mathcal{L}'| \delta,$$

where the first equality follows from De-Morgan's law; the first inequality follows from the union bound; the last inequality is because $\mathbb{P}\left(\mathcal{E}_\ell\right) \geq 1 - \delta$. Setting $\delta = \frac{1}{T|\mathcal{L}'|}$, we obtain

$$\mathbb{P}\left(\sup_{\ell \in \mathcal{L}'} \mathrm{REG}_\ell \leq \underbrace{4\sqrt{KT} + \sqrt{2T \log\left(T |\mathcal{L}'|\right)}}_{=:\Delta}\right) \geq 1 - \frac{1}{T}.$$

Note that, $\mathbb{E}\left[\mathcal{S}\right] = \mathbb{P}(\mathcal{A})\mathbb{E}\left[\mathcal{S}|\mathcal{A}\right] + \mathbb{P}(\mathcal{A}')\mathbb{E}\left[\mathcal{S}|\mathcal{A}'\right]$, where $\mathcal{A}$ denotes the event that $\mathcal{S} \leq \Delta$. Using the facts $\mathbb{E}\left[\mathcal{S}|\mathcal{A}\right] \leq \Delta$, $\mathbb{P}(\mathcal{A}') \leq \frac{1}{T}$, and $\mathbb{E}\left[\mathcal{S}|\mathcal{A}'\right] \leq 2T$ since $\ell \in [-1, 1]$, we have

$$\mathbb{E}\left[\mathcal{S}\right] \leq 2 + \Delta,$$

which completes the proof. $\qquad \square$

Before proceeding further, we first define the notion of cover and covering numbers.

**Definition D.1** (Cover and Covering Number). *The $\epsilon$-cover of a function class $\mathcal{F}$ defined over a domain $\mathcal{X}$ is a function class $\mathcal{C}_\epsilon$ such that, for any $f \in \mathcal{F}$ there exists $g \in \mathcal{C}_\epsilon$ such that $\sup_{x \in \mathcal{X}} |f(x) - g(x)| \leq \epsilon$. The covering number $M(\mathcal{F}, \epsilon; \|.\|_\infty)$ is then defined as $M(\mathcal{F}, \epsilon; \|.\|_\infty) := \min\{|\mathcal{C}_\epsilon| ; \mathcal{C}_\epsilon \text{ is an } \epsilon\text{-cover of } \mathcal{F}\}$, i.e., the size of the minimal cover.*

The $\|.\|_\infty$ in the notation $M(\mathcal{F}, \epsilon; \|.\|_\infty)$ is used to represent the fact that the "distance" between two functions $f, g$ is measured with respect to the $\|.\|_\infty$ norm, i.e., $\sup_{x \in \mathcal{X}} |f(x) - g(x)|$. Such a definition can be generalized to more general distance metrics/pseudo-metrics, but is not required for our purposes. Note that the cover $\mathcal{C}_\epsilon$ in Definiton D.1 is not necessarily a subset of $\mathcal{F}$. We refer to Wainwright (2019) for an exhaustive treatment of cover and covering numbers of different classes $\mathcal{F}$'s.

Let $\mathcal{C}_\epsilon$ be a minimal $\epsilon$-cover of $\mathcal{F}$. For each $g \in \mathcal{C}_\epsilon$, let $\mathcal{S}_{g,\epsilon}$ be the collection of functions $f \in \mathcal{F}$ such that $g$ is a representative of $f$, i.e.,

$$\mathcal{S}_{g,\epsilon} := \left\{f \in \mathcal{F} \mid \sup_{x \in \mathcal{X}} |f(x) - g(x)| \leq \epsilon\right\}. \tag{6}$$

Clearly, $\cup_{g \in \mathcal{C}_\epsilon} \mathcal{S}_{g,\epsilon} = \mathcal{F}$. For each $\mathcal{S}_{g,\epsilon}$, fix a $f_{lead} \in \mathcal{S}_{g,\epsilon}$ (chosen arbitrarily) as a "leader". Let $\mathcal{L}_{lead,\mathcal{F}}$ denote the collection of these leaders. It is clear that $|\mathcal{L}_{lead,\mathcal{F}}| \leq |\mathcal{C}_\epsilon| = M(\mathcal{F}, \epsilon; \|.\|_\infty)$.

In the following lemma, we generalize the result of Lemma D.2 to the case when $\mathcal{L}'$ is a possibly infinite subset of $\mathcal{L}$. Our proof is based on applying the result of Lemma D.2 to the leader set of $\mathcal{L}'$.

**Lemma D.3.** *Fix a subset $\mathcal{L}' \subseteq \mathcal{L}$ with a covering number $M(\mathcal{L}', \epsilon; \|.\|_\infty)$. Then, the sequence of forecasts made by Algorithm 1 satisfies for any $\epsilon > 0$,*

$$\mathsf{UCal}_{\mathcal{L}'} \leq 2 + 4\epsilon T + 4\sqrt{KT} + \sqrt{2T \log\left(T \cdot M(\mathcal{L}', \epsilon; \|.\|_\infty)\right)}.$$

*Proof.* Let $\mathcal{C}_\epsilon$ be an $\epsilon$-cover of $\mathcal{L}'$ of size $M(\mathcal{L}', \epsilon; \|.\|_\infty)$ and $\mathcal{L}_{lead} \subset \mathcal{L}'$ be the corresponding leader set. Applying Lemma D.2, we have

$$\mathbb{E}\left[\sup_{\ell \in \mathcal{L}_{lead}} \mathrm{REG}_\ell\right] \leq 2 + 4\sqrt{KT} + \sqrt{2T \log(T|\mathcal{L}_{lead}|)} \leq 2 + 4\sqrt{KT} + \sqrt{2T \log(T \cdot M(\mathcal{L}', \epsilon; \|.\|_\infty))}.$$

Now, fix any $\ell \in \mathcal{L}'$ and let $g \in \mathcal{C}_\epsilon$ be the representative of $\ell$, and $\ell'$ correspond to the leader of the partition $\mathcal{S}_{g,\epsilon}$ that $\ell$ belongs to. By definition we have the following:

$$|g(\boldsymbol{p}, \boldsymbol{y}) - \ell(\boldsymbol{p}, \boldsymbol{y})| \leq \epsilon, \quad |g(\boldsymbol{p}, \boldsymbol{y}) - \ell'(\boldsymbol{p}, \boldsymbol{y})| \leq \epsilon$$

for all $\boldsymbol{p} \in \Delta_K, \boldsymbol{y} \in \mathcal{E}$. Therefore, it follows from the triangle inequality that $|\ell(\boldsymbol{p}, \boldsymbol{y}) - \ell'(\boldsymbol{p}, \boldsymbol{y})| \leq 2\epsilon$. As usual, let $\boldsymbol{\beta} = \frac{1}{T} \sum_{t=1}^{T} \boldsymbol{y}_t$ denote the empirical average of the outcomes. Then,

$$\ell(\boldsymbol{p}_t, \boldsymbol{y}_t) - \ell(\boldsymbol{\beta}, \boldsymbol{y}_t) \leq \ell'(\boldsymbol{p}_t, \boldsymbol{y}_t) - \ell'(\boldsymbol{\beta}, \boldsymbol{y}_t) + 4\epsilon \implies \mathrm{REG}_\ell \leq \mathrm{REG}_{\ell'} + 4\epsilon T.$$

Taking supremum with respect to $\ell \in \mathcal{L}$ on both sides, followed by expectation, we obtain

$$\mathbb{E}\left[\sup_{\ell \in \mathcal{L}} \mathrm{REG}_\ell\right] \leq \mathbb{E}\left[\sup_{\ell \in \mathcal{L}_{lead}} \mathrm{REG}_\ell\right] + 4\epsilon T \leq 2 + 4\epsilon T + 4\sqrt{KT} + \sqrt{2T \log\left(T \cdot M(\mathcal{L}', \epsilon; \|.\|_\infty)\right)},$$

which finishes the proof. $\qquad\square$

# E Regret of FTL for Locally Lipschitz Functions

**Lemma E.1.** *Suppose that for a loss function $\ell$, there exists a constant $G_{[\frac{1}{T}, 1]}$ such that for each $i \in [K]$, $\ell(\boldsymbol{p}, \boldsymbol{e}_i)$ is locally Lipschitz in the sense that $|\ell(\boldsymbol{p}, \boldsymbol{e}_i) - \ell(\boldsymbol{p}', \boldsymbol{e}_i)| \leq G_{[\frac{1}{T}, 1]} \|\boldsymbol{p} - \boldsymbol{p}'\|$ for all $\boldsymbol{p}, \boldsymbol{p}' \in \Delta_K$ such that $p_i, p_i' \in [\frac{1}{T}, 1]$. Then, the regret of FTL with respect to this loss is at most $2K + G_{[\frac{1}{T}, 1]}(1 + \log T)$.*

*Proof.* Using the Be-the-Leader lemma, we know that the regret of FTL can be bounded as $\mathrm{REG} \leq 2 + \sum_{i=1}^{K} \sum_{t \geq 2; \boldsymbol{y}_t = \boldsymbol{e}_i} \ell(\boldsymbol{p}_t, \boldsymbol{e}_i) - \ell(\boldsymbol{p}_{t+1}, \boldsymbol{e}_i)$. Assume that $\mathcal{E}_m \subseteq \mathcal{E}$ of size $m \leq K$ contains all the outcomes chosen by the adversary over $T$ rounds, i.e., $\boldsymbol{y}_t \in \mathcal{E}_m$ for all $t \in [T]$. For each $\boldsymbol{e}_i \in \mathcal{E}_m$, let $k_i$ denote the total number of time instants $t$ such that $\boldsymbol{y}_t = \boldsymbol{e}_i$ and let $\mathcal{T}_i := \{t_{i,1}, \ldots, t_{i,k_i}\}$ denote those time instants. Then, we have

$$\mathrm{REG}_\ell \leq 2 + \sum_{\boldsymbol{e}_i \in \mathcal{E}_m} \sum_{t \geq 2; \boldsymbol{y}_t = \boldsymbol{e}_i} \ell(\boldsymbol{p}_t, \boldsymbol{e}_i) - \ell(\boldsymbol{p}_{t+1}, \boldsymbol{e}_i)$$

$$\leq 2m + \sum_{\boldsymbol{e}_i \in \mathcal{E}_m} \sum_{t \in \mathcal{T}_i \setminus \{t_{i,1}\}} \ell(\boldsymbol{p}_t, \boldsymbol{e}_i) - \ell(\boldsymbol{p}_{t+1}, \boldsymbol{e}_i),$$

where the last inequality follows by bounding $\ell(\boldsymbol{p}_t, \boldsymbol{e}_i) - \ell(\boldsymbol{p}_{t+1}, \boldsymbol{e}_i)$ with 2 for all the $t$'s where an outcome appears for the first time. For each $\boldsymbol{e}_i \in \mathcal{E}_m$ and for all $t > t_{i,1}$, we have $p_{t,i} = \frac{n_{t-1,i}}{t-1} \geq \frac{1}{T}$. Therefore,

$$\mathrm{REG}_\ell \leq 2m + G_{[\frac{1}{T}, 1]} \sum_{\boldsymbol{e}_i \in \mathcal{E}_m} \sum_{t \in \mathcal{T}_i \setminus \{t_{i,1}\}} \|\boldsymbol{p}_t - \boldsymbol{p}_{t+1}\|$$

$$= 2m + G_{[\frac{1}{T}, 1]} \sum_{\boldsymbol{e}_i \in \mathcal{E}_m} \sum_{t \in \mathcal{T}_i \setminus \{t_{i,1}\}} \left\|\frac{\boldsymbol{n}_{t-1}}{t-1} - \frac{\boldsymbol{n}_t}{t}\right\|.$$

Proceeding similar to the proof of Theorem 3, we can show that

$$\mathrm{REG}_\ell \leq 2m + G_{[\frac{1}{T}, 1]} \sum_{\boldsymbol{e}_i \in \mathcal{E}_m} \sum_{t \in \mathcal{T}_i \setminus \{t_{i,1}\}} \frac{1}{t} = 2m + G_{[\frac{1}{T}, 1]} \left(\sum_{t=1}^{T} \frac{1}{t} - \sum_{\boldsymbol{e}_i \in \mathcal{E}_m} \frac{1}{t_{i,1}}\right)$$

$$\leq 2K + G_{[\frac{1}{T}, 1]}(1 + \log T),$$

where the last inequality follows by dropping the negative term, $m \leq K$, and $\sum_{j=2}^{T} \frac{1}{j} \leq \int_1^T \frac{1}{z} dz = \log T$. This completes the proof. $\qquad\square$

**Example E.1.** *Consider for instance the loss whose univariate form is $\ell(\boldsymbol{p}) = -\tilde{c}_K \sum_{i=1}^{K} p_i^\alpha$, where $\alpha \in (1, 2)$, and $\tilde{c}_K > 0$ is a normalizing constant (which only depends on $K$) to ensure that $\ell(\boldsymbol{p}, \boldsymbol{y})$*

*is bounded in $[-1, 1]$. Clearly, $\ell(\boldsymbol{p})$ is concave and thus the induced loss $\ell(\boldsymbol{p}, \boldsymbol{y})$ is proper as per Lemma 1. For the chosen $\ell(\boldsymbol{p})$, $\ell(\boldsymbol{p}, \boldsymbol{e}_1)$ is given by*

$$\ell(\boldsymbol{p}, \boldsymbol{e}_1) = \ell(\boldsymbol{p}) + (1 - p_1)\nabla_1\ell(\boldsymbol{p}) - \sum_{i=2}^K p_i\nabla_i\ell(\boldsymbol{p}) = -\tilde{c}_K\left((1-\alpha)\sum_{i=1}^K p_i^\alpha + \alpha p_1^{\alpha-1}\right). \quad (7)$$

*It is easy to verify that $\nabla_1\ell(\boldsymbol{p}, \boldsymbol{e}_1) = -\tilde{c}_K \cdot \alpha(\alpha-1)p_i^{\alpha-2}(1-p_1)$ and $\nabla_i\ell(\boldsymbol{p}, \boldsymbol{e}_1) = -\tilde{c}_K \cdot \alpha(1-\alpha)p_i^{\alpha-1}$ for all $i > 1$. Thus, for a large $T$, $G_{[\frac{1}{T}, 1]}$ is of order $\mathcal{O}(\alpha(\alpha-1)\tilde{c}_KT^{2-\alpha})$. This yields a regret bound $\mathcal{O}(K + \alpha(\alpha-1)\tilde{c}_KT^{2-\alpha}\log T)$, which for $\alpha \in (\frac{3}{2}, 2)$ is better (with respect to $T$) than the $\mathcal{O}(\sqrt{KT})$ bound obtained in Theorem 1.*

## F   Proof of Theorem 4

**Theorem 4.** *There exists a proper Lipschitz loss $\ell$ such that: for any algorithm $\mathsf{ALG}$, there exists a choice of $\boldsymbol{y}_1, \ldots, \boldsymbol{y}_T$ by an oblivious adversary such that the expected regret of $\mathsf{ALG}$ is $\Omega(\log T)$.*

As mentioned, the loss we use in this lower bound construction is the squared loss $\ell(\boldsymbol{p}, \boldsymbol{y}) := \|\boldsymbol{p} - \boldsymbol{y}\|^2$ (we ignore the constant $1/2$ here for simplicity, which clearly does not affect the proof). It is clearly convex in $\boldsymbol{p}$ for any $\boldsymbol{y}$. The univariate form of the loss is

$$\ell(\boldsymbol{p}) = \mathbb{E}_{\boldsymbol{y}\sim\boldsymbol{p}}[\ell(\boldsymbol{p}, \boldsymbol{y})] = \sum_{i=1}^K p_i\|\boldsymbol{p} - \boldsymbol{e}_i\|^2 = \|\boldsymbol{p}\|^2 + 1 - 2\sum_{i=1}^K p_i\langle\boldsymbol{p}, \boldsymbol{e}_i\rangle = 1 - \|\boldsymbol{p}\|^2.$$

We now follow the three steps outlined in Section 4.1.

**Step 1:**   First, it is well-known that for convex losses, deterministic algorithms are as powerful as randomized algorithms. Formally, let $\mathcal{A}_{rand}$ and $\mathcal{A}_{det}$ be the class of randomized algorithms and deterministic algorithms respectively for the forecasting problem. Then the following holds:

**Lemma F.1.** *For any loss $\ell(\boldsymbol{p}, \boldsymbol{y})$ that is convex in $\boldsymbol{p} \in \Delta_K$ for any $\boldsymbol{y} \in \mathcal{E}$, we have*

$$\inf_{\mathsf{ALG}\in\mathcal{A}_{rand}} \sup_{\boldsymbol{y}_1,\ldots,\boldsymbol{y}_T\in\mathcal{E}} \mathbb{E}\left[\mathrm{REG}_\ell\right] = \inf_{\mathsf{ALG}\in\mathcal{A}_{det}} \sup_{\boldsymbol{y}_1,\ldots,\boldsymbol{y}_T\in\mathcal{E}} \mathrm{REG}_\ell.$$

*Proof.* The direction "$\leq$" is trivial since $\mathcal{A}_{det} \subseteq \mathcal{A}_{rand}$. For the other direction, it suffices to show that for any randomized algorithm $\mathsf{ALG} \in \mathcal{A}_{rand}$, one can construct a deterministic algorithm $\mathsf{ALG}' \in \mathcal{A}_{det}$ such that, for any fixed sequence $\boldsymbol{y}_1, \ldots, \boldsymbol{y}_T \in \mathcal{E}$, the expected regret of $\mathsf{ALG}$ is lower bounded by the regret of $\mathsf{ALG}'$. To do so, it suffices to let $\mathsf{ALG}'$ output the expectation of the randomized output of $\mathsf{ALG}$ at each time $t$. Since the loss if convex, by Jensen's inequality, the loss of $\mathsf{ALG}'$ is at most the expect loss of $\mathsf{ALG}$. This finishes the proof. $\square$

Since there is no difference between an oblivious adversary and an adaptive adversary for deterministic algorithms, $\inf_{\mathsf{ALG}\in\mathcal{A}_{det}} \sup_{\boldsymbol{y}_1,\ldots,\boldsymbol{y}_T\in\mathcal{E}} \mathrm{REG}_\ell$ can be written as

$$\mathrm{VAL} = \left\langle\!\!\left\langle \inf_{\boldsymbol{p}_t\in\Delta_K} \sup_{\boldsymbol{y}_t\in\mathcal{E}} \right\rangle\!\!\right\rangle_{t=1}^T \left[\sum_{t=1}^T \ell(\boldsymbol{p}_t, \boldsymbol{y}_t) - \inf_{\boldsymbol{p}\in\Delta_K}\sum_{t=1}^T \ell(\boldsymbol{p}, \boldsymbol{y}_t)\right],$$

where $\left\langle\!\langle\inf_{\boldsymbol{p}_t\in\Delta_K} \sup_{\boldsymbol{y}_t\in\mathcal{E}}\rangle\!\right\rangle_{t=1}^T$ is a shorthand for the iterated expression

$$\inf_{\boldsymbol{p}_1\in\Delta_K} \sup_{\boldsymbol{y}_1\in\mathcal{E}} \inf_{\boldsymbol{p}_2\in\Delta_K} \sup_{\boldsymbol{y}_2\in\mathcal{E}} \ldots\ldots \inf_{\boldsymbol{p}_{T-1}\in\Delta_K} \sup_{\boldsymbol{y}_{T-1}\in\mathcal{E}} \inf_{\boldsymbol{p}_T\in\Delta_K} \sup_{\boldsymbol{y}_T\in\mathcal{E}}.$$

Let $\boldsymbol{n} \in \mathbb{N}_{\geq}^K$ be a vector such that $n_i$ represents the cumulative number of the outcome $i$, and $r \in \{0, \ldots, T\}$ represent the number of remaining rounds. For any $\boldsymbol{n}$ and $r$ such that $\|\boldsymbol{n}\|_1 + r = T$, define $\mathcal{V}_{\boldsymbol{n},r}$ recursively as

$$\mathcal{V}_{\boldsymbol{n},r} = \inf_{\boldsymbol{p}\in\Delta_K} \sup_{\boldsymbol{y}\in\mathcal{E}} \mathcal{V}_{\boldsymbol{n}+\boldsymbol{y},r-1} + \ell(\boldsymbol{p}, \boldsymbol{y})$$

with $\mathcal{V}_{\boldsymbol{n},0} = -\inf_{\boldsymbol{p}\in\Delta_K}\sum_{i=1}^K n_i\ell(\boldsymbol{p}, \boldsymbol{e}_i)$. It is then clear that $\mathrm{VAL}$ is simply $\mathcal{V}_{\boldsymbol{0},T}$.

**Step 2:** We proceed to rewrite and simplify $\mathcal{V}_{\boldsymbol{n},r}$ as follows:

$$
\mathcal{V}_{\boldsymbol{n},r} = \inf_{\boldsymbol{p} \in \Delta_K} \sup_{\boldsymbol{y} \in \mathcal{E}} \mathcal{V}_{\boldsymbol{n}+\boldsymbol{y},r-1} + \ell(\boldsymbol{p},\boldsymbol{y}) \tag{8}
$$

$$
= \inf_{\boldsymbol{p} \in \Delta_K} \sup_{\boldsymbol{q} \in \Delta_K} \mathbb{E}_{\boldsymbol{y}\sim\boldsymbol{q}} \left[\mathcal{V}_{\boldsymbol{n}+\boldsymbol{y},r-1} + \ell(\boldsymbol{p},\boldsymbol{y})\right]
$$

$$
= \sup_{\boldsymbol{q} \in \Delta_K} \inf_{\boldsymbol{p} \in \Delta_K} \mathbb{E}_{\boldsymbol{y}\sim\boldsymbol{q}} \left[\mathcal{V}_{\boldsymbol{n}+\boldsymbol{y},r-1} + \ell(\boldsymbol{p},\boldsymbol{y})\right]
$$

$$
= \sup_{\boldsymbol{q} \in \Delta_K} \inf_{\boldsymbol{p} \in \Delta_K} \sum_{i=1}^{K} q_i \mathcal{V}_{\boldsymbol{n}+\boldsymbol{e}_i,r-1} + q_i \ell(\boldsymbol{p},\boldsymbol{e}_i)
$$

$$
= \sup_{\boldsymbol{q} \in \Delta_K} \inf_{\boldsymbol{p} \in \Delta_K} \sum_{i=1}^{K} q_i \mathcal{V}_{\boldsymbol{n}+\boldsymbol{e}_i,r-1} + q_i \left(\ell(\boldsymbol{p}) + \langle \nabla\ell(\boldsymbol{p}), \boldsymbol{e}_i - \boldsymbol{p}\rangle\right)
$$

$$
= \sup_{\boldsymbol{q} \in \Delta_K} \inf_{\boldsymbol{p} \in \Delta_K} \sum_{i=1}^{K} q_i \mathcal{V}_{\boldsymbol{n}+\boldsymbol{e}_i,r-1} + \ell(\boldsymbol{p}) + \langle \nabla\ell(\boldsymbol{p}), \boldsymbol{q} - \boldsymbol{p}\rangle
$$

$$
= \sup_{\boldsymbol{q} \in \Delta_K} \sum_{i=1}^{K} q_i \mathcal{V}_{\boldsymbol{n}+\boldsymbol{e}_i,r-1} + \ell(\boldsymbol{q}),
$$

where the second equality follows since $\mathbb{E}_{\boldsymbol{y}\sim\boldsymbol{q}}\left[\mathcal{V}_{\boldsymbol{n}+\boldsymbol{y},r-1} + \ell(\boldsymbol{p},\boldsymbol{y})\right]$ is a linear function in $\boldsymbol{q}$, and the infimum/supremum of a linear function is attained at the boundary; the third equality follows from the minimax theorem as $\mathbb{E}_{\boldsymbol{y}\sim\boldsymbol{q}}\left[\mathcal{V}_{\boldsymbol{n}+\boldsymbol{y},r-1} + \ell(\boldsymbol{p},\boldsymbol{y})\right]$ is convex in $\boldsymbol{p}$ and concave in $\boldsymbol{q}$; the final equality is because $\ell(\boldsymbol{p}) + \langle \nabla\ell(\boldsymbol{p}), \boldsymbol{q} - \boldsymbol{p}\rangle \geq \ell(\boldsymbol{q})$ which follows from the concavity of the univariate form of $\ell$, and equality is attained at $\boldsymbol{p} = \boldsymbol{q}$.

Throughout the subsequent discussion, we consider $K = 2$ and use the concrete form of the squared loss. Writing $\mathcal{V}_{(n_1,n_2),r}$ as $\mathcal{V}_{n_1,n_2,r}$ for simplicity, we simplify the recurrence to

$$
\mathcal{V}_{n_1,n_2,r} = \sup_{\boldsymbol{q} \in \Delta_2} q_1 \mathcal{V}_{n_1+1,n_2,r-1} + q_2 \mathcal{V}_{n_1,n_2+1,r-1} + 1 - q_1^2 - q_2^2
$$

$$
= \sup_{q \in [0,1]} \mathcal{V}_2 + (\mathcal{V}_1 - \mathcal{V}_2)q - 2(q^2 - q),
$$

where $\mathcal{V}_1, \mathcal{V}_2$ is a shorthand for $\mathcal{V}_{n_1+1,n_2,r-1}, \mathcal{V}_{n_1,n_2+1,r-1}$ respectively. It is straightforward to show via the KKT conditions that

$$
\sup_{q \in [0,1]} \mathcal{V}_2 + (\mathcal{V}_1 - \mathcal{V}_2)q - 2(q^2 - q) = \begin{cases} \mathcal{V}_2 & \text{if } \mathcal{V}_1 - \mathcal{V}_2 < -2, \\ \frac{(\mathcal{V}_1-\mathcal{V}_2)^2}{8} + \frac{\mathcal{V}_1+\mathcal{V}_2}{2} + \frac{1}{2} & \text{if } -2 \leq \mathcal{V}_1 - \mathcal{V}_2 \leq 2, \quad (\mathcal{R}) \\ \mathcal{V}_1 & \text{if } \mathcal{V}_1 - \mathcal{V}_2 > 2. \end{cases}
$$

Thus, $(\mathcal{R})$ (with $\mathcal{V}_1, \mathcal{V}_2$ replaced by $\mathcal{V}_{n_1+1,n_2,r-1}, \mathcal{V}_{n_1,n_2+1,r-1}$) represents the recurrence we wish to solve for to obtain $\mathcal{V}_{0,0,T}$. The base case of $(\mathcal{R})$ is the following:

$$
\mathcal{V}_{n,T-n,0} = -T\ell\left(\left[\frac{n}{T}, 1 - \frac{n}{T}\right]\right) = 2T \cdot \frac{n}{T} \cdot \left(\frac{n}{T} - 1\right) \tag{9}
$$

which holds for all $n$ such that $0 \leq n \leq T$. In the next lemma we show that $-2 \leq \mathcal{V}_1 - \mathcal{V}_2 \leq 2$, therefore, it is sufficient to solve the recursion $(\mathcal{R})$ corresponding to this case only.

**Lemma F.2.** *The recurrence $(\mathcal{R})$ is also equal to the following:*

$$
\mathcal{V}_{n_1,n_2,r} = \frac{(\mathcal{V}_{n_1+1,n_2,r-1} - \mathcal{V}_{n_1,n_2+1,r-1})^2}{8} + \frac{\mathcal{V}_{n_1+1,n_2,r-1} + \mathcal{V}_{n_1,n_2+1,r-1}}{2} + \frac{1}{2}. \tag{$\mathcal{R}'$}
$$

*Proof.* It suffices to show that the condition $|\mathcal{V}_{n_1+1,n_2,r-1} - \mathcal{V}_{n_1,n_2+1,r-1}| \leq 2$ always holds. Rewriting $n_2 + 1$ as $n_2$ and $r - 1$ as $r$, this is equivalent to showing

$$
|\mathcal{V}_{n_1,n_2,r} - \mathcal{V}_{n_1+1,n_2-1,r}| \leq 2 \tag{10}
$$

for all $n_1, n_2, r$ such that $n_1 + n_2 + r = T$, and the arguments in (10) are well defined, i.e., $0 \leq n_1 \leq T - r - 1$, and $1 \leq n_2 \leq T - r$. We prove this by an induction on $r$.

**Base Case:** This corresponds to $r = 0$. For $0 \le n \le T - 1$, $|\mathcal{V}_{n,T-n,0} - \mathcal{V}_{n+1,T-n-1,0}|$ is equal to

$$2T \cdot \left| \frac{n}{T} \cdot \left( \frac{n}{T} - 1 \right) - \frac{n+1}{T} \cdot \left( \frac{n+1}{T} - 1 \right) \right| = 2 \cdot \left| 1 - \frac{2n+1}{T} \right| \le 2 \cdot \left( 1 - \frac{1}{T} \right) \le 2,$$

which verifies the base case.

**Induction Hypothesis:** Fix a $k \in \{1, \dots, T\}$; assume that (10) holds for $r = k - 1$.

**Induction Step:** We show that (10) holds for $r = k$. It follows from $(\mathcal{R})$ and the induction hypothesis that

$$\mathcal{V}_{n_1,n_2,k} = \frac{1}{8} \left( \mathcal{V}_{n_1+1,n_2,k-1} - \mathcal{V}_{n_1,n_2+1,k-1} \right)^2 + \frac{\mathcal{V}_{n_1+1,n_2,k-1} + \mathcal{V}_{n_1,n_2+1,k-1}}{2} + \frac{1}{2},$$

$$\mathcal{V}_{n_1+1,n_2-1,k} = \frac{1}{8} \left( \mathcal{V}_{n_1+2,n_2-1,k-1} - \mathcal{V}_{n_1+1,n_2,k-1} \right)^2 + \frac{\mathcal{V}_{n_1+2,n_2-1,k-1} + \mathcal{V}_{n_1+1,n_2,k-1}}{2} + \frac{1}{2}.$$

Let $\alpha_1 := \mathcal{V}_{n_1+2,n_2-1,k-1}, \alpha_2 := \mathcal{V}_{n_1+1,n_2,k-1}, \alpha_3 := \mathcal{V}_{n_1,n_2+1,k-1}, \Delta := \mathcal{V}_{n_1+1,n_2-1,k} - \mathcal{V}_{n_1,n_2,k}$. Subtracting the equations above and expressing in terms of the defined quantities, we obtain

$$\Delta = \frac{\alpha_1 + \alpha_2}{2} + \frac{(\alpha_1 - \alpha_2)^2}{8} - \frac{\alpha_2 + \alpha_3}{2} - \frac{(\alpha_2 - \alpha_3)^2}{8}$$

$$= \frac{\alpha_1 - \alpha_3}{2} + \frac{(\alpha_1 - \alpha_3) \cdot (\alpha_1 + \alpha_3 - 2\alpha_2)}{8}$$

$$= \frac{x + y}{2} + \frac{(x + y) \cdot (x - y)}{8}$$

$$= \frac{(x + 2)^2 - (y - 2)^2}{8},$$

where we have defined $x := \alpha_1 - \alpha_2, y := \alpha_2 - \alpha_3$. It follows from the induction hypothesis that $|x| \le 2, |y| \le 2$, therefore $|\Delta| \le 2$. Summarizing, we have shown that $|\mathcal{V}_{n_1+1,n_2-1,k} - \mathcal{V}_{n_1,n_2,k}| \le 2$ which completes the proof via induction. $\qquad \square$

**Step 3:** Since $n_1 + n_2 + r = T$, we may express $\mathcal{V}_{n_1,n_2,r}$ in terms of $n_1$, $n_2$, and $T$. In the next lemma, we show that $\mathcal{V}_{n_1,n_2,r}$ exhibits a very special structure when expressed in this manner; this allows us to reduce $\mathcal{V}_{0,0,T}$ to solving a one dimensional recurrence.

**Lemma F.3.** *For all $n_1, n_2$ such that $n_1, n_2 \ge 0$, and $n_1 + n_2 = T - r$, it holds that*

$$\mathcal{V}_{n_1,n_2,r} = \frac{(n_1 - n_2)^2}{2} \cdot u_r - \frac{2n_1 n_2}{T} + v_r, \tag{11}$$

*where $\{u_r\}_{r=0}^{T}, \{v_r\}_{r=0}^{T}$ are sequences that depend only on $T$, and are defined by the following recurrences:*

$$u_{r+1} = u_r + \left( u_r + \frac{1}{T} \right)^2, \quad v_{r+1} = \frac{u_r}{2} + v_r + \frac{r+1}{T} - \frac{1}{2} \tag{12}$$

*for all $0 \le r \le T - 1$, with $u_0 = v_0 = 0$.*

*Proof.* Similar to Lemma F.2, the proof shall follow by an induction on $r$.

**Base Case:** For $r = 0$, it follows from (9) that $\mathcal{V}_{n_1,n_2,0} = -\frac{2n_1 n_2}{T}$, which is consistent with (11) and $u_0 = v_0 = 0$.

**Induction Hypothesis:** Fix a $k \in \{0, \dots, T - 1\}$. Assume that (11) holds for $r = k$.

**Induction Step:** We show that (11) holds for $r = k + 1$. From Lemma F.2, we have

$$\mathcal{V}_{n_1,n_2,k+1} = \frac{(\mathcal{V}_{n_1+1,n_2,k} - \mathcal{V}_{n_1,n_2+1,k})^2}{8} + \frac{\mathcal{V}_{n_1+1,n_2,k} + \mathcal{V}_{n_1,n_2+1,k}}{2} + \frac{1}{2}. \tag{13}$$

It follows from the induction hypothesis that

$$\mathcal{V}_{n_1+1,n_2,k} = \frac{(n_1 + 1 - n_2)^2}{2} \cdot u_k - \frac{2(n_1 + 1) \cdot n_2}{T} + v_k,$$

$$\mathcal{V}_{n_1,n_2+1,k} = \frac{(n_1 - n_2 - 1)^2}{2} \cdot u_k - \frac{2n_1 \cdot (n_2 + 1)}{T} + v_k.$$

Define $\delta := \mathcal{V}_{n_1+1,n_2,k} - \mathcal{V}_{n_1,n_2+1,k}$ and $\sigma := \mathcal{V}_{n_1+1,n_2,k} + \mathcal{V}_{n_1,n_2+1,k}$. Subtracting the equations above, we obtain

$$\delta = \frac{(n_1 + 1 - n_2)^2 - (n_1 - n_2 - 1)^2}{2} \cdot u_k + \frac{2n_1 \cdot (n_2 + 1) - 2(n_1 + 1) \cdot n_2}{T}$$

$$= 2(n_1 - n_2) \cdot \left( u_k + \frac{1}{T} \right).$$

Adding the equations above, we obtain

$$\sigma = \frac{(n_1 + 1 - n_2)^2 + (n_1 - n_2 - 1)^2}{2} \cdot u_k - \frac{2n_1 \cdot (n_2 + 1) + 2(n_1 + 1) \cdot n_2}{T} + 2v_k$$

$$= \left( (n_1 - n_2)^2 + 1 \right) \cdot u_k - \frac{4n_1 n_2}{T} - \frac{2(n_1 + n_2)}{T} + 2v_k$$

$$= (n_1 - n_2)^2 \cdot u_k - \frac{4n_1 n_2}{T} + u_k + 2v_k + \frac{2(r + 1)}{T} - 2,$$

where the last equality follows since $n_1 + n_2 = T - k - 1$. Expressing $\mathcal{V}_{n_1,n_2,k+1}$ in terms of $\delta, \sigma$, we have $\mathcal{V}_{n_1,n_2,k+1} = \frac{\delta^2}{8} + \frac{\sigma}{2} + \frac{1}{2}$. Substituting $\delta, \sigma$, we obtain

$$\mathcal{V}_{n_1,n_2,k+1} = \frac{(n_1 - n_2)^2}{2} \cdot \left( u_k + \left( u_k + \frac{1}{T} \right)^2 \right) - \frac{2n_1 n_2}{T} + \frac{u_k}{2} + v_k + \frac{(r + 1)}{T} - \frac{1}{2}$$

$$= \frac{(n_1 - n_2)^2}{2} \cdot u_{k+1} - \frac{2n_1 n_2}{T} + v_{k+1},$$

which completes the induction step. The proof is hence complete by induction. $\qquad\square$

Since $\mathrm{VAL} = \mathcal{V}_{0,0,T} = v_T$, it only remains to bound $v_T$. Note that the recursion describing $v$ is coupled with $u$. However, since we only want to bound $v_T$, we can sum the recursion describing $v$ to obtain

$$\sum_{r=0}^{T-1} (v_{r+1} - v_r) = \frac{1}{2} \cdot \sum_{r=0}^{T-1} u_r + \frac{1}{T} \cdot \sum_{r=0}^{T-1} (r + 1) - \frac{T}{2} = \frac{1}{2} \cdot \left( \sum_{r=0}^{T-1} u_r + 1 \right).$$

Moreover, since the summation with respect to $v$ telescopes and $v_0 = 0$, we have

$$v_T = \frac{1}{2} \cdot \left( \sum_{r=0}^{T-1} u_r + 1 \right).$$

Therefore, it remains to bound $\sum_{r=0}^{T-1} u_r$. Define $a_r := u_r + \frac{1}{T}$ so that $v_T = \frac{1}{2} \sum_{r=0}^{T-1} a_r$. The recurrence (12) describing $u$ reduces to $a_{r+1} = a_r + a_r^2$ for all $0 \le r \le T - 1$, with $a_0 = \frac{1}{T}$. In the next result, we obtain bounds on $a_r$.

**Lemma F.4.** *For all $0 \le r \le T - 1$, it holds that $a_r \le \frac{1}{T-r}$.*

*Proof.* As usual, the proof shall follow by an induction on $r$. Since $a_0 = \frac{1}{T}$, the base case is trivially satisfied. Fix a $k \in \{0, \ldots, T - 2\}$, and assume that $a_k \le \frac{1}{T-k}$. Since $a_{k+1} = a_k + a_k^2$, we have

$$a_{k+1} \le \frac{1}{(T - k)^2} + \frac{1}{T - k} = \frac{T - k + 1}{(T - k)^2} = \frac{(T - k)^2 - 1}{(T - k)^2} \cdot \frac{1}{T - k - 1} \le \frac{1}{T - k - 1}.$$

This completes the induction step. $\qquad\square$

**Lemma F.5.** *For all $0 \le r \le T - 1$, it holds that $a_r \ge \frac{1}{T-r+\log T}$. Furthermore,*

$$\sum_{r=0}^{T-1} a_r \ge \log \left( \frac{T}{\log T + 1} + 1 \right) = \Omega(\log T).$$

*Proof.* According to Lemma F.4, we can write $a_r$ as $\frac{1}{T-r+b_r}$ for some non-negative sequence $\{b_r\}$ with $b_0 = 0$. We next obtain the recurrence describing $\{b_r\}$. In particular, since $a_{r+1} = a_r(a_r + 1)$, we have

$$\frac{1}{T - (r+1) + b_{r+1}} = \frac{1}{T - r + b_r} \cdot \left( \frac{1}{T - r + b_r} + 1 \right),$$

which on simplifying (by multiplying both sides with $(T - (r+1) + b_{r+1})(T - r + b_r)$) yields

$$b_{r+1} = b_r + \frac{1 + b_r - b_{r+1}}{T - r + b_r}. \tag{14}$$

Next, we shall show by an induction on $r$ that $b_r \leq \sum_{i=0}^{r-1} \frac{1}{T-i}$ for all $0 \leq r \leq T - 1$. Since $b_0 = 0$, the base case is trivially satisfied. Fix a $k \in \{0, \ldots, T-2\}$ and assume that $b_k \leq \sum_{i=0}^{k-1} \frac{1}{T-i}$. We consider two cases depending on whether or not $b_{k+1} \geq b_k$. If $b_{k+1} < b_k$, the induction step holds trivially. If $b_{k+1} \geq b_k$, it follows from the recurrence (14) that

$$b_{k+1} \leq b_k + \frac{1}{T - k + b_k} \leq b_k + \frac{1}{T-k} \leq \sum_{i=0}^{k} \frac{1}{T-i},$$

which completes the induction step. Therefore, we have established that $b_r \leq \sum_{i=0}^{r-1} \frac{1}{T-i}$ for all $0 \leq r \leq T - 1$. It then follows that

$$b_r \leq \sum_{i=0}^{T-2} \frac{1}{T-i} \leq \int_1^T \frac{dz}{z} = \log T,$$

for all $0 \leq r \leq T - 1$, which translates to $a_r \geq \frac{1}{T-r+\log T}$. This completes the proof of the first part of the lemma. With this lower bound on $a_r$, we can lower bound $\sum_{r=0}^{T-1} a_r$ as

$$\sum_{r=0}^{T-1} a_r \geq \sum_{r=0}^{T-1} \frac{1}{T - r + \log T} = \sum_{i=1}^{T} \frac{1}{i + \log T} \geq \int_1^{T+1} \frac{dz}{z + \log T} = \log\left( \frac{T}{\log T + 1} + 1 \right),$$

which is $\Omega(\log T)$ for a large $T$. This completes the proof. $\qquad\square$

To conclude, we have shown

$$\text{VAL} = \mathcal{V}_{0,0,T} = v_T = \frac{1}{2} \left( \sum_{r=0}^{T-1} u_r + 1 \right) = \frac{1}{2} \sum_{r=0}^{T-1} a_r = \Omega(\log T),$$

proving Theorem 4.

## G    Proof of Theorem 5

**Theorem 5.** *The regret of FTL for learning any $\ell \in \mathcal{L}_{dec}$ is at most $2K + (K+1)\beta_\ell(1 + \log T)$ for some universal constant $\beta_\ell$ which only depends on $\ell$ and $K$. Consequently, FTL ensures $\mathsf{PUCal}_{\mathcal{L}_{dec}} = \mathsf{UCal}_{\mathcal{L}_{dec}} = \mathcal{O}((\sup_{\ell \in \mathcal{L}_{dec}} \beta_\ell)K \log T)$.*

*Proof.* We work with the notation established in the proof of Lemma E.1. The regret of FTL can be bounded as

$$\text{REG} \leq 2m + \sum_{e_i \in \mathcal{E}_m} \sum_{t \in \mathcal{T}_i \setminus \{t_{i,1}\}} \underbrace{\ell(p_t, e_i) - \ell(p_{t+1}, e_i)}_{\delta_{t,i}}.$$

In the subsequent steps, we shall bound $\delta_{t,i}$. We begin in the following manner:

$$\begin{aligned}
\delta_{t,i} &= \ell(p_t) + \langle e_i - p_t, \nabla\ell(p_t) \rangle - \ell(p_{t+1}) - \langle e_i - p_{t+1}, \nabla\ell(p_{t+1}) \rangle \\
&= \ell(p_t) - \ell(p_{t+1}) + [\nabla\ell(p_t)]_i - [\nabla\ell(p_{t+1})]_i + \langle p_{t+1}, \nabla\ell(p_{t+1}) \rangle - \langle p_t, \nabla\ell(p_t) \rangle \\
&\leq [\nabla\ell(p_t)]_i - [\nabla\ell(p_{t+1})]_i + \langle p_t, \nabla\ell(p_{t+1}) - \nabla\ell(p_t) \rangle,
\end{aligned}$$

where the first equality follows from Lemma 1; the first inequality follows since $\ell(\boldsymbol{p}_t) \leq \ell(\boldsymbol{p}_{t+1}) + \langle \nabla \ell(\boldsymbol{p}_{t+1}), \boldsymbol{p}_t - \boldsymbol{p}_{t+1} \rangle$ which follows from the concavity of $\ell$. Next, by the Mean Value Theorem,

$$\nabla \ell(\boldsymbol{p}_{t+1}) - \nabla \ell(\boldsymbol{p}_t) = \nabla^2 \ell(\boldsymbol{p}_t + v(\boldsymbol{p}_{t+1} - \boldsymbol{p}_t)) \cdot (\boldsymbol{p}_{t+1} - \boldsymbol{p}_t)$$

for some $v \in [0, 1]$. Note that $p_{t,i} = \frac{n_{t-1,i}}{t-1}, p_{t+1,i} = \frac{n_{t-1,i}+1}{t}$ (since $\boldsymbol{y}_t = \boldsymbol{e}_i$), therefore $p_{t+1,i} \geq p_{t,i}$. Let $\boldsymbol{\xi}_t := \boldsymbol{p}_t + v(\boldsymbol{p}_{t+1} - \boldsymbol{p}_t)$. Then,

$$[\nabla \ell(\boldsymbol{p}_t)]_i - [\nabla \ell(\boldsymbol{p}_{t+1})]_i = \langle [\nabla^2 \ell(\boldsymbol{\xi}_t)]_i, \boldsymbol{p}_t - \boldsymbol{p}_{t+1} \rangle,$$

where $[\nabla^2 \ell(\boldsymbol{\xi}_t)]_i$ denotes the $i$-th row of $[\nabla^2 \ell(\boldsymbol{\xi}_t)]$; we arrive at

$$\delta_{t,i} \leq \langle [\nabla^2 \ell(\boldsymbol{\xi}_t)]_i, \boldsymbol{p}_t - \boldsymbol{p}_{t+1} \rangle + \langle \boldsymbol{p}_{t+1} - \boldsymbol{p}_t, \nabla^2 \ell(\boldsymbol{\xi}_t) \boldsymbol{p}_t \rangle$$

$$= |\nabla^2_{i,i} \ell(\boldsymbol{\xi}_t)| \cdot (p_{t+1,i} - p_{t,i}) + \sum_{j=1}^K p_{t,j} \cdot \nabla^2_{j,j} \ell(\boldsymbol{\xi}_t) \cdot (p_{t+1,j} - p_{t,j})$$

$$\leq |\nabla^2_{i,i} \ell(\boldsymbol{\xi}_t)| \cdot (p_{t+1,i} - p_{t,i}) + \sum_{j \neq i} p_{t,j} \cdot \nabla^2_{j,j} \ell(\boldsymbol{\xi}_t) \cdot (p_{t+1,j} - p_{t,j})$$

$$\leq \sum_{j=1}^K |\nabla^2_{j,j} \ell(\boldsymbol{\xi}_t)| \cdot |(p_{t,j} - p_{t+1,j})|, \tag{15}$$

where the first equality follows since $p_{t+1,i} \geq p_{t,i}$ and $\nabla^2_{i,i} \ell(\boldsymbol{\xi}_t) \leq 0$; the second inequality follows by dropping the term $p_{t,i} \cdot \nabla^2_{i,i} \ell(\boldsymbol{\xi}_t) \cdot (p_{t+1,i} - p_{t,i})$ which is non-positive; the final inequality is because, for $j \neq i$, we have $p_{t+1,j} = \frac{n_{t-1,j}}{t}$ and $p_{t,j} = \frac{n_{t-1,j}}{t-1}$, therefore $p_{t+1,j} \leq p_{t,j}$, and bounding $p_{t,j} \leq 1$.

To proceed with the further bounding, we apply Lemma 2 and utilize the growth condition on the Hessian $|\nabla^2_{j,j} \ell(\boldsymbol{\xi}_t)| \leq \tilde{c}_K \cdot c_j \cdot \max\left(\frac{1}{\xi_{t,j}}, \frac{1}{1-\xi_{t,j}}\right)$ for some constant $c_j > 0$ and all $j \in [K]$ (here $\tilde{c}_K$ is the scaling constant such that $\ell(\boldsymbol{p}) = \tilde{c}_K \sum_{i=1}^K \ell_i(p_i)$). Let $\beta_{\ell,j} := \tilde{c}_K \cdot c_j$. Using the bound on $|\nabla^2_{j,j} \ell(\boldsymbol{\xi}_t)|$ to bound the term in (15), we obtain

$$\delta_{t,i} \leq \sum_{j=1}^K \beta_{\ell,j} \cdot \max\left(\frac{1}{\xi_{t,j}}, \frac{1}{1-\xi_{t,j}}\right) \cdot |p_{t,j} - p_{t+1,j}|. \tag{16}$$

Next, we shall bound the terms $\mathcal{T}_1 := \frac{p_{t+1,i} - p_{t,i}}{\xi_{t,i}}, \mathcal{T}_2 := \frac{p_{t+1,i} - p_{t,i}}{1 - \xi_{t,i}}, \mathcal{T}_3 := \frac{p_{t,j} - p_{t+1,j}}{\xi_{t,j}}$, and $\mathcal{T}_4 := \frac{p_{t,j} - p_{t+1,j}}{1 - \xi_{t,j}}$, where the index $j$ in $\mathcal{T}_3$ and $\mathcal{T}_4$ runs over $j \neq i$. Note that,

$$\mathcal{T}_1 = \frac{p_{t+1,i} - p_{t,i}}{p_{t,i} + v(p_{t+1,i} - p_{t,i})} \leq \left(\frac{p_{t+1,i} - p_{t,i}}{p_{t,i}}\right) = \left(\frac{n_{t-1,i}+1}{n_{t-1,i}} \cdot \frac{t-1}{t} - 1\right) \leq \frac{1}{n_{t-1,i}},$$

where the first equality follows from the definition of $\boldsymbol{\xi}_t$; the first inequality follows since $p_{t+1,i} \geq p_{t,i}$ and $v \in [0, 1]$. Similarly,

$$\mathcal{T}_2 = \frac{p_{t+1,i} - p_{t,i}}{1 - p_{t,i} - v(p_{t+1,i} - p_{t,i})} \leq \frac{p_{t+1,i} - p_{t,i}}{1 - p_{t+1,i}}$$

$$= \left(\frac{n_{t-1,i}+1}{t} - \frac{n_{t-1,i}}{t-1}\right) \cdot \frac{1}{1 - \frac{n_{t-i,i}+1}{t}}$$

$$= \frac{t - 1 - n_{t-1,i}}{t \cdot (t-1)} \cdot \frac{t}{t - 1 - n_{t-1,i}} = \frac{1}{t-1},$$

where the first inequality follows since $p_{t+1,i} \geq p_{t,i}$. For $\mathcal{T}_3$, we have

$$\mathcal{T}_3 = \frac{1 - \frac{p_{t+1,j}}{p_{t,j}}}{1 + v\left(\frac{p_{t+1,j}}{p_{t,j}} - 1\right)} = \frac{\frac{1}{t}}{1 - \frac{v}{t}} \leq \frac{1}{t-1},$$

where the inequality follows since $v \in [0, 1]$. Finally, we bound $\mathcal{T}_4$ as

$$\mathcal{T}_4 = \frac{1 - \frac{p_{t+1,j}}{p_{t,j}}}{\frac{1}{p_{t,j}} - \left(1 + v\left(\frac{p_{t+1,j}}{p_{t,j}} - 1\right)\right)} = \frac{\frac{1}{t}}{\frac{t-1}{n_{t-1,j}} - 1 + \frac{v}{t}} \leq \frac{n_{t-1,j}}{t} \cdot \frac{1}{t - 1 - n_{t-1,j}} \leq \frac{1}{n_{t-1,i}},$$

where the final inequality is because $\sum_{j=1}^{K} n_{t-1,j} = t - 1$. Collecting the bounds on $\mathcal{T}_1, \mathcal{T}_2, \mathcal{T}_3$, and $\mathcal{T}_4$, and substituting them back in (16), we get

$$\delta_{t,i} \leq \sum_{j=1}^{K} \beta_{\ell,j} \cdot \max\left(\frac{1}{n_{t-1,i}}, \frac{1}{t-1}\right) \leq \sum_{j=1}^{K} \beta_{\ell,j} \cdot \left(\frac{1}{n_{t-1,i}} + \frac{1}{t-1}\right).$$

Summing over all $t$, we obtain $\text{REG}_\ell \leq 2m + \beta_\ell(\mathcal{S}_1 + \mathcal{S}_2)$, where $\mathcal{S}_1 := \sum_{e_i \in \mathcal{E}_m} \sum_{t \in \mathcal{T}_i \setminus \{t_{i,1}\}} \frac{1}{n_{t-1,i}}$, $\mathcal{S}_2 := \sum_{e_i \in \mathcal{E}_m} \sum_{t \in \mathcal{T}_i \setminus \{t_{i,1}\}} \frac{1}{t-1}$, and $\beta_\ell := \sum_{i=1}^{K} \beta_{\ell,i}$. Note that the subscript in $\beta_\ell$ denotes that the constant is dependent on the loss $\ell$ and only depends on $\ell$ and $K$. We bound $\mathcal{S}_1$ as

$$\mathcal{S}_1 = \sum_{e_i \in \mathcal{E}_m} \sum_{j=1}^{k_i - 1} \frac{1}{j} \leq m \sum_{t=1}^{T} \frac{1}{j} \leq m(1 + \log T) \leq K(1 + \log T).$$

Next, note that $\mathcal{S}_2 \leq \sum_{t=1}^{T} \frac{1}{t} \leq 1 + \log T$. Thus, $\mathcal{S}_1 + \mathcal{S}_2 \leq (K + 1)(1 + \log T)$, which yields $\text{REG}_\ell \leq 2K + (K + 1)\beta_\ell(1 + \log T)$. This completes the proof. $\square$

## H  Proof of Lemma 2

**Lemma 2.** *For a function $f$ that is concave, Lipschitz, and bounded over $[0, 1]$ and twice continuously differentiable over $(0, 1)$, there exists a constant $c > 0$ such that $|f''(p)| \leq c \cdot \max\left(\frac{1}{p}, \frac{1}{1-p}\right)$ for all $p \in (0, 1)$.*

*Proof.* Since $f$ is twice differentiable, if $|f''(p)|$ does not approach infinity at the boundary of $[0, 1]$, then there is nothing to prove. In the rest of the proof, we assume that $|f''(p)|$ approaches infinity both when $p$ approaches $0$ and when $p$ approaches $1$ (the case when it only approaches infinty at one side is exactly the same).

Using a technical result from Lemma H.1, there exists an $\epsilon_0 \in (0, 1)$ such that for any $p \in (0, \epsilon_0]$,

$$|f''(p)| = \frac{|f'(q) - f'(0)|}{q}$$

for some $q \in [p, 1]$, which can be further bounded by

$$\frac{|f'(q) - f'(0)|}{p} \leq \frac{|f'(q)| + |f'(0)|}{p} \leq 2 \cdot \frac{\sup_{q \in [0,1]} |f'(q)|}{p} \leq c_1 \max\left(\frac{1}{p}, \frac{1}{1-p}\right)$$

with $c_1 := 2 \sup_{q \in [0,1]} |f'(q)|$ (finite due to $f$ being Lipschitz). Similarly, there exists an $\epsilon_1 \in (0, 1)$ such that for any $p \in [1 - \epsilon_1, 1)$,

$$|f''(p)| = \frac{|f'(1) - f'(q)|}{1 - q}$$

for some $q \in [0, p]$, which is further bounded by

$$\frac{|f'(1) - f'(q)|}{1 - p} \leq \frac{|f'(1)| + |f'(q)|}{1 - p} \leq c_1 \max\left(\frac{1}{p}, \frac{1}{1-p}\right).$$

Finally, for any $p \in (\epsilon_0, 1 - \epsilon_1)$, we trivially bound $|f''(p)|$ as

$$|f''(p)| \leq \max\left(\frac{1}{p}, \frac{1}{1-p}\right) \underbrace{\sup_{q \in (\epsilon_0, 1-\epsilon_1)} \frac{|f''(q)|}{\max\left(\frac{1}{q}, \frac{1}{1-q}\right)}}_{c_2},$$

where $c_2$ is finite since $f$ is twice continuously differentiable in $(0, 1)$. Setting $c = \max(c_1, c_2)$ finishes the proof. $\square$

**Lemma H.1.** *Let $f$ satisfy the conditions of Lemma 2 and additionally $\lim_{p \to 0^+} |f''(p)| = \infty$ and $\lim_{p \to 1^-} |f''(p)| = \infty$. Then there exists $\epsilon_0 \in (0, 1)$ such that for any $p \in (0, \epsilon_0]$, we have $f'(q) - f'(0) = f''(p)q$ for some $q \in [p, 1]$. Similarly, there exists $\epsilon_1 \in (0, 1)$ such that for any $p \in [1 - \epsilon_1, 1)$, we have $f'(1) - f'(q) = f''(p)(1 - q)$ for some $q \in [0, p]$.*

*Proof.* For simplicity, we only prove the first part of the lemma since the proof of the second part follows the same argument. Since $f$ is twice continuously differentiable in $(0, 1)$ and $\lim_{p \to 0^+} |f''(p)| = \infty$, there exists an $\epsilon \in (0, 1)$ such that $|f''|$ is decreasing in $(0, \epsilon]$. Since $f$ is concave, this is equivalent to $f''$ being increasing in $(0, \epsilon]$.

Now, applying Mean Value Theorem, we know that there exists $\epsilon_0 \in (0, \epsilon]$ such that $f'(\epsilon) - f'(0) = f''(\epsilon_0)\epsilon$. It remains to prove that for any $p \in (0, \epsilon_0]$, the function $g(q) := f'(q) - f'(0) - f''(p)q$ has a root in $[p, 1]$. This is true because

$$g(p) = f'(p) - f'(0) - f''(p)p = (f''(\xi) - f''(p))p \le 0,$$

where the equality is by Mean Value Theorem again for some point $\xi \in [0, p]$ and the inequality holds since $f''$ is increasing in $(0, \epsilon]$. On the other hand, we have

$$g(\epsilon) = f'(\epsilon) - f'(0) - f''(p)\epsilon = (f''(\epsilon_0) - f''(p))\epsilon \ge 0,$$

where the equality holds by the definition of $\epsilon_0$ and the inequality holds since again $f''$ is increasing in $(0, \epsilon]$. Applying the Intermediate Value Theorem then shows that $g(q)$ has a root in $[p, \epsilon]$, which finishes the proof. $\square$

