# OpenReview forum: "Optimal Multiclass U-Calibration Error and Beyond"
_NeurIPS.cc/2024/Conference — NeurIPS 2024 poster_

### Official Review · Reviewer_jtgy · 2024-07-12

**Soundness:** 4
**Presentation:** 4
**Contribution:** 3
**Rating:** 7
**Confidence:** 5

**Summary:**

This paper studies online forecasting algorithms for achieving multiclass U-calibration. In the standard setting of online forecasting, a forecaster must produce a prediction p_t each day for an event with K possible outcomes (p_t is a distribution over K outcomes). The event then occurs (with outcome x_t), and the forecaster must suffer some loss L(p_t, x_t). The forecaster wants to minimize their total loss over all T rounds (or alternatively, their regret compared to the best fixed prediction in hindsight).

Traditionally, L is chosen to be a specific proper scoring rule (e.g., the quadratic loss). But you may wish to produce predictions that are good for any proper scoring rule L (and hence that induce low regret for any downstream agent). The U-calibration error (introduced by Kleinberg et al. in an earlier paper) is the maximum regret of the forecaster with respect to some bounded proper scoring rule. Kleinberg et al. constructed a randomized online forecaster that achieves O(K * sqrt(T)) U-calibration error for the K outcome case and posed an open question of whether this is tight.

This paper does the following:
1. They provide an algorithm that achieves O(sqrt(KT)) U-calibration error, and proves that this is tight (in fact, that there is a single proper scoring rule where any forecaster must incur Omega(sqrt(KT)) regret).
2. They then show that there are specific subclasses of loss functions where it is possible to get much stronger U-calibration bounds (for the modified definition of U-calibration only looking at these losses). E.g, for Lipschitz-bounded proper scoring rules and decomposable proper scoring rules, it is possible to get O(log T) U-calibration bounds. Also worth noting is that these bounds hold for a slightly stronger form of U-calibration (the previous bounds mentioned are all for “pseudo U-calibration”, whereas these are for actual U-calibration, the difference being swapping the order of taking the max over loss functions and taking the expectation over the algorithm’s randomness).

Kleinberg et al.’s randomized forecaster is based on a modification of follow-the-perturbed-leader to the forecasting setting. The optimal O(sqrt(KT)) algorithm presented in this paper also builds off this FTPL approach, but uses a different perturbation distribution (geometric instead of uniform) to achieve sqrt(K). The stronger bounds follow from showing that the deterministic Follow-The-Leader algorithm works for these subclasses of losses.

**Strengths:**

Evaluation:

Multiclass prediction is an important problem where you often want guarantees for downstream agents. The standard solution to this -- calibration -- is notoriously hard to achieve in multiclass settings (e.g., in the online setting calibration bounds for d-dimensional outcomes often scale as O(T^{1 - 1/(d+2)}), so it is important to understand alternate guarantees (such as U-calibration) which are tractable but guarantee the same outcome. This paper almost entirely resolves a very natural open problem about U-calibration understanding the optimal U-calibration rates for multiclass calibration (the only thing left open really being a somewhat technical distinction of what is possible for “pseudo-U-calibration” vs “U-calibration”). As such, I found this paper very interesting and would expect it to be of interest to many NeurIPS attendees interested in online learning and/or calibration.

One possible criticism of this paper is that it doesn’t do too much that it is entirely novel from a technical perspective -- e.g., the main result follows from an observation that an FTPL variant analyzed by Daskalakis and Syrgkanis in 2016 can be immediately applied to the FTPL forecaster introduced by Kleinberg et al. to get the O(sqrt(KT)) calibration result. But personally I think that this is a very nice observation (and it is actually nice that the result is so simple). Similarly, it is not too surprising that FTL is good for some subclass of losses (as FTL is known to get logarithmic regret for e.g. strongly convex OCO), but I also think this is a very nice observation to point out.

**Weaknesses:**

See previous section.

**Questions:**

No specific questions, feel free to reply to anything in the review.

**Limitations:**

Authors have adequately addressed limitations.

---

> ### Author Rebuttal · Authors · 2024-08-05
>
> We sincerely thank you for your comments and evaluation of the manuscript. About your comment on the technical perspective, we would like to highlight a result that seems to be overlooked by the reviewer (since it is not mentioned in the summary of the review): our Theorem 4, an $\Omega(\log T)$ lower bound for any algorithm when learning with the squared loss, is the most technical part of our manuscript and requires substantially new ideas. As we point out after Theorem 4, while squared loss is known to admit $\Theta(\log T)$ regret in other online learning problems such as that from Abernethy et al. (2008), as far as we know there is no study on our setting where the decision set is the simplex and the adversary has only finite choices. We believe that this result is particularly original and significant.

---

> > ### Comment · Reviewer_jtgy · 2024-08-12
> >
> > Thank you for your response! I have read the other reviews and comments and maintain my (positive) evaluation.

---

### Official Review · Reviewer_Uz2D · 2024-07-13

**Soundness:** 3
**Presentation:** 3
**Contribution:** 2
**Rating:** 6
**Confidence:** 3

**Summary:**

This paper studies the calibrating for multiclass distribution forecasting while considering all proper functions simultaneously and contributes minimax optimal errors for a variety of settings.

**Strengths:**

Originality:
The work is a novel combination of known techniques, especially Kleinberg et al. (2023).
It is clarified how this work differs from previous contributions.
The related work is somewhat adequately cited.

Quality:
The submission technically sound.
The claims are generally well supported by theoretical analysis.
The methods used are appropriate.
This is a complete piece of work answering a recent open problem.
The authors are more or less careful and honest about evaluating their work, and placing it in the literature.

Clarity:
The submission is clearly written and well organized.

Significance:
The results are important in the sense that it conclusively generates optimal approaches for a multitude of scenarios.
Other researchers and practitioners are likely to use the ideas and build on them, as this work did, possibly for integration into machine learning applications.
It advances the state of the art in a demonstrable way through theoretical analysis, performance bounds and open problem answers.
It provides unique conclusions about existing methods and some unique theoretical approaches.

**Weaknesses:**

Originality:
The tasks or methods do not really stand out as new, more emphasis what the work introduces would help.

Clarity:
It is a bit lacking at times in adequately informing the reader regarding the contents, especially towards the end (pages 8 and 9).

Significance:
The difficulty of the task the submission addresses (demonstrably in a better way than the previous works) is questionable.

**Questions:**

Major Questions:
- Page 2 Line 42: isn't p from a continuum, how is a sum over p defined?
- Page 5 Line 208: how is (a) claimed? Explain.

Minor Questions:
- Page 2 Line 39: what do you insert instead of dropped notations?
- Page 8 Line 322: why 2K?

Suggestions:
- Page 6 Line 228: check grammar.
- Page 7 Line 280: missing comma.

**Limitations:**

The authors have adequately addressed the limitations.

---

> ### Author Rebuttal · Authors · 2024-08-05
>
> We sincerely thank you for your comments and evaluation of the manuscript. Regarding the weakness in originality and significance, we would like to highlight that our Theorem 4, an $\Omega(\log T)$ lower bound for any algorithm when learning with the squared loss, is the most technical part of our manuscript and requires substantially new ideas. As we point out after Theorem 4, while squared loss is known to admit $\Theta(\log T)$ regret in other online learning problems such as that from Abernethy et al. (2008), as far as we know there is no study on our setting where the decision set is the simplex and the adversary has only finite choices. We believe that this result is particularly original and significant.
>
> Your other questions are addressed below:
>
> **Major Questions:**
> > Page 2 Line 42: isn't p from a continuum, how is a sum over p defined?
>
> Note that even though $p$ is from a continuum, there are at most $T$ non-zero summands in this summation since there are at most $T$ different forecasts ($p_{t}$). This is why the notation $\sum_{p\in \Delta_K}$ is well defined and in fact standard in the calibration literature.
>
> > Page 5 Line 208: how is (a) claimed? Explain.
>
> The exact reasoning can be found in Lines 474-481, but the intuitive explanation is simply that by the properness of the V-shaped loss, it is straightforward to see that predicting a uniform distribution at each time $t$ is in expectation the optimal choice for the learner in this environment (where the outcome is also uniform over $\{e_1, \ldots, e_K\}$), and this optimal strategy has exactly 0 loss.
>
> **Minor Questions:**
>
> > Page 2 Line 39: what do you insert instead of dropped notations?
>
> As we mention in Lines 38 and 39, whenever the subscript is dropped, both UCal and PUCal are to be thought of with respect to the class of all proper losses, for which we use the notation $\mathcal{L}$. Whenever a different subscript $\mathcal{L}’$ appears, both quantities are defined with respect to that specific class of losses, which would be clear from the context.
>
> > Page 8 Line 322: why $2K$?
>
> At every time when a new label is chosen by the adversary, we bound the regret trivially by $2$ since $\ell$ is bounded in $[-1, 1]$. Since the number of distinct labels is $K$, this contributes at most $2K$ to the overall regret.
>
> **Suggestions**: Thank you for the suggestions. We shall do the needful in the subsequent revisions.

---

> > ### Comment · Reviewer_Uz2D · 2024-08-13
> >
> > Thank you for your response. I have read and considered it, as well as the other reviews and rebuttals. My opinion about the paper is still towards acceptance.

---

### Official Review · Reviewer_PchL · 2024-07-13

**Soundness:** 4
**Presentation:** 4
**Contribution:** 4
**Rating:** 8
**Confidence:** 3

**Summary:**

This paper closes an open problem regarding U calibration, left by Kleinberg et al. (2023). It is shown that a modified version of Kleinberg et al's algorithm recovers a classical FTPL algorithm of Daskalakis and Syrgkanis (2016), which improves the pseudo U calibration error in Kleinberg et al (2023) from $O(K\sqrt{T})$ to the optimal rate $O(\sqrt{KT})$. Then, the paper considers several special classes of proper losses, and shows that FTL guarantees $O(\log T)$ U calibration error. Finally, it is shown that although FTL works in such special cases, FTPL is necessary in general.

**Strengths:**

This paper is a very solid and comprehensive contribution to the theory of U calibration. It answers an open problem left from an already amazing prior work, and complements the existing generic theory by several interesting special cases. Although the proposed algorithm is a small modification from Kleinberg et al. (2023), the established equivalence to the classical results of Daskalakis and Syrgkanis is novel. The intuition and the analysis are both quite natural. Various extensions are thoroughly analyzed, and the presentation is exceptionally clear.

**Weaknesses:**

This is one of the rare cases where finding a weakness is hard. One thing I might say is the lack of operational impact. Although U calibration is a relatively new framework, the intuition of the proper losses and the fact that FTL & FTPL work well somewhat suggest that it is more of a new way to look at existing online learning algorithms, rather than a new framework to design different online learning algorithms. This will arguably limit the impact of such results in the broader ML community, which is my reason for not giving the paper an even higher score.

However, I would say such practical limitation is quite common in recent learning theory papers. Based on the technical quality, this paper is still a very good contribution to learning theory.

**Questions:**

The follow up questions I'm interested in are already discussed by the authors as future directions. The discussed directions there make sense.

**Limitations:**

Societal impact not applicable due to the theoretical nature.

---

> ### Author Rebuttal · Authors · 2024-08-05
>
> We sincerely thank you for your comments and evaluation of the manuscript.

---

### Official Review · Reviewer_aUK6 · 2024-07-13

**Soundness:** 4
**Presentation:** 4
**Contribution:** 4
**Rating:** 7
**Confidence:** 3

**Summary:**

This works considered the the problem of making sequential non-contextual probabilistic predictions over $K$ classes with low U-calibration error.
The authors improved the upper bounds for U-calibration error from $O(K\sqrt{T})$ to $O(\sqrt{KT})$ after $T$ rounds, and they showed an existing algorithm, Follow-the-Perturbed-Leader (FTPL), achieves this upper bound.
The also proved a matching lower bound, and shows the optimality of FTPL.
Further, they provided strengthened bounds of $\Theta(\log T)$ under various additional assumptions on the proper losses.

**Strengths:**

- This paper is well written and the logic is easy to follow.
- This paper makes original and significant contribution by improving the upper bound, providing an algorithm, and proves a matching lower bound. In a sense, this paper solves the problem of online probabilistic multiclass predictions with U-calibration error.
- The mathematical development is sound and rigorous.

**Weaknesses:**

No outstanding weakness. Typos and questions are given under questions.

**Questions:**

- L25: Based on the regret definition, it seems the it can be negative because the best prediction in hindsight is fixed for all time steps? For example, an oracle $p = \arg\min_p \sum_{t=1}^{T} \ell(p, y_t)$ achieves 0 regret, and an oracle  $p_t = y_t$ for all $t$ has negative regret.
- L124: is the condition "if and only if"? See the interpretation immediately after Theorem 2 Gneiting and Raftery (2007): "Phrased slightly differently, a regular scoring rule S is proper if and only if the expected score function G(p) = S(p,p) is convex on Pm".
- In Section 3.1, would it be nice to accompany the algorithm description and theoretical guarantees by some intuition why this randomized algorithm works? For readers not familiar with FTPL, a naive forecast would be to just predict the class frequency $\boldsymbol{\beta}$ up to the current time step. Why is it worse than FTPL and what is the intuition behind the randomization in the algorithm?
- In Section 3.1, is it better to use the notation system established in previous sections to translate the FTPL algorithm in Daskalakis and Syrgkanis (2016)? For example, I think both are fine, just curious which one is more accepted.
- L181: dose -> does

**Limitations:**

Limitations are addressed theoretically in Section 4.3.

---

> ### Author Rebuttal · Authors · 2024-08-05
>
> We sincerely thank you for your comments and evaluation of the manuscript. Your questions are addressed below:
>
> >L25: Based on the regret definition, it seems the it can be negative because the best prediction in hindsight is fixed for all time steps? For example, an oracle $p = \text{arg}\min_p\sum_{t = 1} ^ {T} \ell(p, y_{t})$ achieves 0 regret, and an oracle $p_{t} = y_{t}$ for all $t$ has negative regret.
>
> Yes, you are correct. The regret can potentially be negative (which is true for most online learning problems), but that only happens when the adversary is very weak.
>
> >L124: is the condition "if and only if"? See the interpretation immediately after Theorem 2 Gneiting and Raftery (2007): "Phrased slightly differently, a regular scoring rule $S$ is proper if and only if the expected score function $G(p) = S(p,p)$ is convex on $\mathcal{P}_m$".
>
> That’s right. The condition is “if and only if”. Although line 124 conveys one direction, the subsequent lines (125, 126) combined with 124 convey the “if and only if” part.
>
> >In Section 3.1, would it be nice to accompany the algorithm description and theoretical guarantees by some intuition why this randomized algorithm works? For readers not familiar with FTPL, a naive forecast would be to just predict the class frequency $\beta$ up to the current time step. Why is it worse than FTPL and what is the intuition behind the randomization in the algorithm?
>
> The naive forecast you mentioned is exactly FTL, which we show suffers linear regret for a certain V-shaped loss in Theorem 6. From its proof, one can see that the intuition is that FTL is unstable in the sense that $\ell(p_t, y_t) - \ell(p_{t+1}, y_t)$ can be as bad as $\Omega(1)$. On the other hand, by introducing randomness, FTPL stabilizes the algorithm and avoids this issue.
>
> >In Section 3.1, is it better to use the notation system established in previous sections to translate the FTPL algorithm in Daskalakis and Syrgkanis (2016)? For example, I think both are fine, just curious which one is more accepted.
>
> We are not sure what you meant, unfortunately. We have indeed followed the approach of introducing the FTPL algorithm in Daskalaskis and Syrgkanis (2016) using their notations and then mapping the notations to our context. Please let us know what is the other usage of notation that you want to compare this to.
>
> >L181: dose -> does
>
> Thank you for pointing out the typo. We shall correct this in the subsequent revisions.

---

### Decision · Program_Chairs · 2024-09-25

**Decision:**

Accept (poster)

**Comment:**

This paper studies the recently developed setting of multiclass U-Calibration (which is a metric that is weaker in some sense than the classical calibration metric, but still has many implications of interest in forecasting, prediction and decision making, and can be achieved at a faster rate in $$T$$ than calibration), and provides the optimal rate in the number of classes $$K$$ through a novel connection to the FTPL algorithm of Daskalakis and Syrgkanis (2016) for the upper bound along with a matching lower bound. Adaptive, faster rates are also provided for special cases (Lipschitz proper losses, decomposable proper losses, low covering number, etc.). All the reviewers are very enthusiastic about this contribution, and I wholeheartedly share their opinion -- this is an important problem that is solved through a remarkably simple connection.